# WHAT EXACTLY DOES GUIDANCE DO IN MASKED DISCRETE DIFFUSION MODELS

**Ye He, Kevin Rojas, Molei Tao**
School of Mathematics
Georgia Institute of Technology
Atlanta, GA 30332, USA
{yhe367,kevin.rojas,mtao}@gatech.edu

## ABSTRACT

Masked discrete diffusion models have been gaining popularity recently, and classifier-free guidance, just like its continuous counterpart, has been proposed to enable efficacious conditional generation by discrete diffusion. To quantify the precise effect of discrete guidance, this article considers masked discrete diffusion with arbitrary data distribution in low dimension, so that the distribution that guided masked discrete diffusion samples from, as well as the sampling dynamics, can be analytically and exactly quantified and interpreted. When the full data distribution is a mixture over classes and the goal is to sample from a specific class, guidance amplifies class-specific regions while suppresses regions shared with other classes. This effect depends on the guidance strength $w$ and induces distinct covariance structures in the sampled distribution. Notably, we observe quantitatively different behaviors in 1D and 2D. We also show that for large $w$, the decay rate of the total variation (TV) along the reverse dynamics is double-exponential in $w$ for both 1D and 2D. These findings highlight the role of guidance, not just in shaping the output distribution, but also in controlling the dynamics of the sampling trajectory. Our theoretical analysis is supported by experiments that illustrate the geometric effects of guidance and its impact on convergence.

## 1 INTRODUCTION

Diffusion models have become an influential tool for generative modeling, offering a flexible framework that performs well across a range of data types including images, audio, and text (Dhariwal & Nichol, 2021; Kong et al., 2021; Li et al., 2022; Ho et al., 2022). Originally formulated in continuous state spaces (Ho et al., 2020; Song et al., 2021), these models simulate a stochastic differential equation that gradually adds Gaussian noise and then learns a reverse process to denoise and reconstruct the data. More recently, discrete diffusion models have been proposed (Campbell et al., 2022; Lou et al., 2023), replacing Gaussian corruption with categorical transitions or masking, which makes them well-suited for language modeling, molecule generation, and protein design (Lou et al., 2023; Nie et al.; Huang et al., 2023; Gruver et al., 2023).

A key innovation that has enhanced the performance and flexibility of diffusion models is guidance. In continuous models, classifier guidance (Dhariwal & Nichol, 2021) and classifier-free guidance (CFG) (Ho & Salimans, 2021; Nichol et al., 2022) steer the reverse process toward desired conditions, such as class labels or text prompts. These methods have been critical to the success of models such as GLIDE (Nichol et al., 2022) and Imagen (Saharia et al., 2022), and theoretical work has begun to explain their mechanism in simplified continuous settings (Bradley & Nakkiran, 2024; Wu et al., 2024; Chidambaram et al., 2024). However, the extension of guidance to discrete diffusion models is much less understood. Recent proposals adapt CFG to discrete domains by modifying transition rates or reweighting transition kernels (Nisonoff et al., 2024; Schiff et al., 2024), and empirical results show notable gains in sample quality and controllability (Schiff et al., 2024; Xiong et al., 2025). However, the theoretical understanding of how guidance affects the dynamics of the diffusion process in discrete state spaces remains limited.

Motivated by this gap, our paper develops a rigorous and quantitative framework for analyzing the effects of CFG in discrete diffusion models introduced in Nisonoff et al. (2024), with a particular focus on masked discrete diffusion models (Campbell et al., 2022; Shi et al., 2024; Sahoo et al., 2024; Ou et al., 2024)—a common subclass of discrete diffusion models. To make the problem tractable while still capturing essential behaviors, we assume exact scores and exact reverse dynamics, and we investigate the following two fundamental questions in low-dimensional settings (1D and 2D):

**Q1.** *How does guidance affect the distribution of the generated samples?*

For this question, we assume that the data distribution $p$ is a mixture of different class distributions (Assumption 1.1). We find that guidance amplifies probability mass in class-specific regions while reducing mass in overlapping regions, which vanish entirely as the guidance strength $w$ grows. The strength of this amplification and reduction depends explicitly on $w$, and the covariance structure of the resulting distribution differs between 1D and 2D.

**Q2.** *How does guidance affect the rate of convergence of the reverse dynamics?*

To address this, we analyze the total variation distance between the distribution along the reverse dynamics and the final sampled distribution. For both 1D and 2D, we show that the decay of this distance exhibits a double-exponential dependence on the guidance strength $w$ when $w \gg 1$.

By characterizing these two aspects—distributional shifts and convergence rates—our work bridges the gap between practical heuristics and theoretical understanding in discrete diffusion with CFG.

**Assumption 1.1.** *Let $\{z_k\}_{k=1}^M$ be the set of $M$ labels, each of which is associated with a class distribution $p(\cdot|z_k)$ supported on $\mathcal{X}_k \subsetneq S$. The full data distribution $p$ is a mixture of distributions $\{p(\cdot|z_k)\}_{k=1}^M$ with weights $\{a_k\}_{k=1}^M$, i.e., $p(\cdot) = \sum_{k=1}^M a_k p(\cdot|z_k)$.*

**Comparison to Existing Work.** Our analysis is closely related to recent theoretical studies of guidance in continuous diffusion models (Bradley & Nakkiran, 2024; Wu et al., 2024; Chidambaram et al., 2024). A key insight from this line of work is that in continuous settings with data distributions such as 1D Gaussian (Bradley & Nakkiran, 2024) or 1D Gaussian/compact-support mixtures (Chidambaram et al., 2024), CFG reshapes the reverse dynamics in a way that makes the generated distribution deviate from the tilted distribution, i.e., the distribution defined by the steered score that emphasizes the conditioning signal. These results, however, rely heavily on Gaussian or compact-support assumptions and are limited to 1D continuous space. By contrast, our study provides the first rigorous analysis of CFG in discrete diffusion models under much more general conditions, where the data distribution can be any finite mixture of arbitrary class distributions. Leveraging the tractability of masked discrete diffusion, we derive explicit reverse dynamics in both 1D and 2D that reveal striking similarities and differences to the continuous case. In particular, we find that in 1D the generated distribution coincides exactly with the tilted distribution—unlike in continuous diffusion—while in 2D deviations do appear, but in a form that we can quantify explicitly rather than only approximately. Moreover, we show that sample diversity decreases as masses in overlapping regions vanish under strong guidance, and that convergence rates exhibit a double-exponential dependence on the guidance parameter, echoing but also amplifying phenomena observed in continuous settings.

**Paper Organization.** The remainder of the paper is organized as follows. Section 2 introduces preliminaries on discrete diffusion models relevant to our analysis. Section 3 presents our theoretical analysis and their implications of the guided diffusion process, and Section 4 provides numerical examples supporting our findings. Conclusions are discussed in Section 5. Additional related work and technical details are included in the Appendix.

## 2 PRELIMINARIES

### 2.1 NOTATIONS

For any $x \in \mathbb{R}^D$ and $A \subset \{1, 2, \cdots, D\}$, we use $x_A \in \mathbb{R}^{|A|}$ to denote restriction of $x$ to coordinates in $A$. $\backslash i$ is used to denote $\{1, 2, \cdots, D\} \setminus \{i\}$. For any distribution $p$, $p(x_A)$ denotes the $A$-marginal of $p$ evaluated at $x_A$. For functions $f, g$, we write $f(w) \sim g(w)$ if $\lim_{w \to \infty} f(w)/g(w) = 1$ and $f(w) = \Theta(g(w))$ if $c_1 g(w) \leq f(w) \leq c_2 g(w)$ for some $c_1, c_2, w_0 > 0$ and all $w > w_0$. For any set $\mathcal{X} \subset \{1, 2, \cdots, N\}^D$ and $1 \leq d \leq D$, $\mathcal{X}_d := \{x_d \mid x \in \mathcal{X}\}$ is the projection of $\mathcal{X}$ on dimension-$d$.

## 2.2 DISCRETE DIFFUSION MODELS

We consider the state space $S = \{1, 2, \cdots, N\}^D$. The data distribution $p$ is represented as a vector in $\mathbb{R}^{N^D}$ that sums up to 1. The discrete diffusion forward process is defined as a continuous-time Markov process (Campbell et al., 2022; Lou et al., 2023), given by the differential equation

$$\frac{\mathrm{d}p_t}{\mathrm{d}t} = Q_t p_t, \quad p_0 = p, \tag{1}$$

where $Q_t \in \mathbb{R}^{N^D \times N^D}$ is the transition rate matrix for all $t \geq 0$ s.t. (1) $Q(y, x) \geq 0$ for all $x, y \in S$ and $x \neq y$; (2) $\sum_{y \in S} Q_t(y, x) = 0$ for all $x \in S$. In this paper, we focus on a widely used effective forward process, the absorbing forward process (Austin et al., 2021; Lou et al., 2023; Shi et al., 2024; Sahoo et al., 2024; Ou et al., 2024), which independently transforms across all dimensions until arrive at the masked state $[M] := (N, N \cdots, N)^\intercal$. The explicit expression of the masked transition rate matrix and its properties will be discussed in Appendix B. The reverse process in discrete diffusion models is defined by

$$\frac{\mathrm{d}q_t}{\mathrm{d}t} = \bar{Q}_{T-t} q_t, \quad q_0 = p_T, \tag{2}$$

where the $\{\bar{Q}_t\}_{0 \leq t \leq T}$ is a sequence of reverse transition rate matrices given by

$$\bar{Q}_t(y, x) = \begin{cases} \frac{p_t(y)}{p_t(x)} Q_t(x, y), & y \neq x, \\ -\sum_{s \neq x} \bar{Q}_t(s, x), & y = x, \end{cases} \tag{3}$$

and $\{p_t\}_{t \geq 0}$ solves equation 1. Equation 2 is the exact reverse of equation 1, i.e., $q_t = p_{T-t}$ for all $t \in [0, T]$. The ratios $\{\frac{p_t(y)}{p_t(x)}\}_{y \in S}$ are the concrete scores (Meng et al., 2022) which generalize the score function $\nabla \log p_t(x)$ in continuous diffusion models. In practice, the concrete scores are learned via denoising entropy matching (Lou et al., 2023) by minimizing the following denoising score entropy:

$$\mathcal{L}_{\mathrm{DSE}} = \mathbb{E}_{x_0 \sim p} \mathbb{E}_{x \sim p_{t|0}(\cdot|x_0)} \Big[ \sum_{y \neq x} s_t^\theta(x, y) - \frac{p_{t|0}(y|x_0)}{p_{t|0}(x|x_0)} \log s_t^\theta(x, y) \Big], \tag{4}$$

where $s_t^\theta(x, y)$ is the parametrized score to approximate $\frac{p_t(y)}{p_t(x)}$. Last, samples are generated approximating the following reverse process:

$$\frac{\mathrm{d}q_t^\theta}{\mathrm{d}t} = \bar{Q}_{T-t}^\theta q_t^\theta, \quad q_0^\theta = \delta_{[M]}, \tag{5}$$

where $\bar{Q}_t^\theta$ replaces the exact concrete score $\frac{p_t(y)}{p_t(x)}$ with $s_t^\theta(x, y)$. Simulation of equation 5 can be performed with the Gillespie's Algorithm (Gillespie, 1976), Tau-leaping (Gillespie, 2001; Campbell et al., 2022) and uniformization (Grassmann, 1977; Chen & Ying, 2024), etc.

In this paper, we assume exact score and exact simulation of equation 5, and focus on the generation ability along the continuous-time reverse dynamics with CFG. The effects of score approximation and numerical discretization are left for future work.

## 2.3 DISCRETE DIFFUSION MODELS WITH CFG

Classifier-free guidance (CFG) is designed to steer the generative process toward samples consistent with a target condition, such as a specific class label. In our paper, we consider to generate from one label class $z$ from the full label classes $\{z_i\}_{i=1}^M$ defined in Assumption 1.1. In the discrete setting, Nisonoff et al. (2024) introduced CFG by tilting the reverse dynamics. Formally, the construction begins with the tilted distribution

$$p^{z,w}(\cdot) \propto p(\cdot) p(z|\cdot)^{1+w} \propto p(\cdot)^{-w} p(\cdot|z)^{1+w}, \tag{6}$$

where $w \geq -1$ is the guidance parameter. This expression makes the effect of guidance transparent: setting $w = -1$ recovers the full data distribution; $w = 0$ yields the conditional distribution on class

$z$; larger positive values of $w$ progressively amplify the likelihood $p(z|x)$, biasing the distribution toward states that are more consistent with class $z$.

Since diffusion models proceed through dynamics, we can't sample from $p^{z,w}$ directly. In continuous diffusion models, one steers the scores $\nabla \log p_t(x)$ during the reverse process. In discrete models, the analogue is to steer the reverse transition rates, thereby embedding the effect of tilting directly into the evolution of the process.

Concretely, alongside the unguided reverse rates $\bar{Q}_t$ in equation 3, we define class-conditional reverse rates $\bar{Q}_t^z$ by evolving the conditional distribution $p(\cdot|z)$ under the same forward transition $Q_t$:

$$\frac{\mathrm{d}p_t(\cdot|z)}{\mathrm{d}t} = Q_t p_t(\cdot|z), \quad p_0 = p(\cdot|z), \tag{7}$$

leading to the conditional reverse rate matrix

$$\bar{Q}_t^z(y,x) = \begin{cases} \frac{p_t(y|z)}{p_t(x|z)} Q_t(x,y), & y \neq x, \\ -\sum_{s \neq x} \bar{Q}_t^z(s,x), & y = x. \end{cases} \tag{8}$$

By analogy with the tilted distribution equation 6, the guided reverse dynamics interpolate between $\bar{Q}_t$ and $\bar{Q}_t^z$ through the CFG rate matrix:

$$\hat{Q}_t^{z,w}(y,x) = \begin{cases} \bar{Q}_t(y,x)^{-w} \bar{Q}_t^z(y,x)^{1+w}, & y \neq x, \\ -\sum_{s \neq x} \hat{Q}_t^{z,w}(s,x), & y = x, \end{cases} \tag{9}$$

with the corresponding evolution

$$\frac{\mathrm{d}q_t^{z,w}}{\mathrm{d}t} = \hat{Q}_{T-t}^{z,w} q_t^{z,w}, \quad q_0^{z,w} = \delta_{[M]}. \tag{10}$$

The choice of guidance strength $w$ can significantly affect the generation:

- when $w = -1$, $\hat{Q}_t^{z,-1} = \bar{Q}_t$, it recovers the unguided reverse process. This generates the entire mixture $p$, spreading probability mass across all classes and offering no control over which class is realized.

- when $w = 0$, $\hat{Q}_t^{z,0} = \bar{Q}_t^z$, it recovers the class-conditional reverse process. This generates exactly from $p(\cdot|z)$, isolating the desired component of the mixture.

Empirically, the best generation results are obtained for intermediate values $w > 0$. In this regime, the process in equation 10 interpolates between reproducing the full mixture and enforcing strict conditioning, often yielding sharper and more faithful samples than either extreme. Yet why this works so well is far from obvious: CFG with $w > 0$ does not correspond to sampling from any simple or explicit distribution, but rather modifies the reverse dynamics in a nonlinear way. A central goal of this paper is to make this phenomenon precise, by rigorously characterizing how guidance reshapes distributions and convergence rates in discrete diffusion models.

## 3   ANALYSIS OF MASKED DIFFUSION WITH CFG IN LOW DIMENSIONS

We now analyze how classifier-free guidance (CFG) reshapes the reverse dynamics in masked discrete diffusion. Our focus is on the low-dimensional settings $D = 1$ and $D = 2$, where explicit formulas for the reverse dynamics can be derived. These tractable cases provide two key insights:

1. **Generated distributions:** the effect of CFG on the final generated distribution relative to the tilted distribution in equation 6.

2. **Convergence rates:** the speed at which the reverse dynamics approach the generated distribution.

Interestingly, the behavior of differs sharply between 1D and 2D: in 1D the generated distribution coincides exactly with the tilted distribution, while in 2D discrepancies appear that we can nonetheless characterize explicitly.

For clarity, we focus on sampling from class $z_1$, denoted by $z$ when the subscript is not essential.

## 3.1 $D = 1$: SINGLE-TOKEN GENERATION

For $D = 1$, the reverse transition rate matrix with CFG simplifies dramatically. It coincide with the reverse rate matrix associated with the tilted distribution $p^{z,w}$, up to a normalization constant $\mathcal{Z}^{z,w} := \sum_{x=1}^{N-1} p(x)^{-w} p(x|z)^{1+w}$. In other words, the guided reverse dynamics in 1D behaves exactly like an unguided reverse dynamics targeting the tilted distribution. For details, please see Proposition B.2.

This leads to the following explicit characterization.

**Theorem 3.1** (1D revserse dynamics). *If $D = 1$ and $q_t^{z,w}$ satisfies the sampling dynamics equation 10, we have that for all $0 \le t \le T$,*

$$
q_t^{z,w}(x) = \begin{cases} \left(1 - \left(\frac{1-e^{-(T-t)}}{1-e^{-T}}\right)^{\mathcal{Z}}\right) p^{z,w}(x), & x = 1, 2, \cdots, N-1, \\ \left(\frac{1-e^{-(T-t)}}{1-e^{-T}}\right)^{\mathcal{Z}}, & x = N, \end{cases} \tag{11}
$$

*where $\mathcal{Z} = \mathcal{Z}^{z,w} = \sum_{x=1}^{N-1} p(x)^{-w} p(x|z)^{1+w}$. Moreover, the generated distribution is exactly the tilted distribution, i.e., $q_T^{z,w} = p^{z,w}$.*

The explicit formula in Theorem 3.1 immediately yields properties of the sampled distribution and convergence rates, whose proofs deferred to Appendix D.

**Proposition 3.1** (1D generated distribution). *Assume the full distribution satisfies Assumption 1.1, depending on the supports of class $z_1$ and other classes, the generated distribution $q_T^{z_1,w}$ admits the following different behaviors:*

*(1) if $\mathcal{X}_1 \cap \mathcal{X}_k = \emptyset$ for all $k \ge 2$, $q_T^{z_1,w} = p(\cdot|z_1)$ for all $w \ge 0$.*

*(2) if $S_1 := \mathcal{X}_1 \cap \left( \cup_{k=2}^M \mathcal{X}_k \right) \neq \emptyset$ and $I_1 := \{k : \mathcal{X}_k \cap \mathcal{X}_1 \neq \emptyset\}$,*

$$
q_T^{z_1,w}(x) \propto \begin{cases} p(x|z_1), & x \in \mathcal{X}_1 \setminus S_1, \\ \left(\frac{a_1 p(x|z_1)}{\sum_{k \in I_1} a_k p(x|z_k)}\right)^w p(x|z_1) & x \in S_1, \\ 0, & otherwise. \end{cases}
$$

*Moreover, as $w \to \infty$, $q_T^{z_1,w} \to p_{\mathcal{X}_1 \setminus S_1}(\cdot|z_1)$ pointwisely, where $p_{\mathcal{X}_1 \setminus S_1}(\cdot|z_1)$ is the restriction of $p(\cdot|z_1)$ to the set $\mathcal{X}_1 \setminus S_1$.*

**Remark 3.1** (Local mean/variance preservation). *Proposition 3.1 suggests that the generated distribution preserves the local mean/ covariance of class-1 distribution within the class-unique region $(\mathcal{X}_1 \setminus S_1)$. Intuitively, CFG in 1D transforms mass in the ambiguous region $S_1$ and redistribute it to class-unique region while preserving the local mean and variance within the unique region. Please refer to Appendix D for the detailed argument.*

**Proposition 3.2** (1D convergence rate). *Under the conditions in Theorem 3.1, for all $0 \le t \le T$ and $w > 0$, $\mathrm{TV}(q_t^{z,w}, p^{z,w}) = \left(\frac{1-e^{-(T-t)}}{1-e^{-T}}\right)^{\mathcal{Z}}$.*

**Remark 3.2** (Double-exponential effect). *The decay of $\mathrm{TV}$ is exponentially in time, with rate $\mathcal{Z}$. For all $w > 0$, alternatively we can represent $\mathcal{Z} = \exp(w\mathcal{D}_{1+w}(p(\cdot|z)\|p))$, where $\mathcal{D}_\alpha(\mu_1|\mu_2) := \frac{1}{\alpha-1} \log \left( \sum_x \frac{\mu_1(x)^\alpha}{\mu_2(x)^{\alpha-1}} \right)$ is the $\alpha$-divergence from $\mu_1$ to $\mu_2$ for all $\alpha \in (0, \infty) \setminus \{1\}$. According to the property of $\alpha$-divergence, we immediately have $\log(\mathcal{Z}^{z,w}) \sim w \sup_x \frac{p(x|z)}{p(x)}$ for $w \gg 1$. Therefore, the overall decay rate of $\mathrm{TV}$ exhibits a double-exponential dependency on $w$. This explains the extremely sharp transitions we observe in practice when $w$ is large.*

## 3.2 $D = 2$: MULTIPLE-TOKEN GENERATION

In 2D, the story changes. The guided reverse rate matrix no longer coincide with that of the tilted distribution (denoted as $\bar{Q}_t[p^{z,w}]$), i.e., $\hat{Q}_t^{z,w} \neq C\bar{Q}_t[p^{z,w}]$ for any constant $C$. This discrepancy marks the departure between the discrete diffusion with CFG in 2D and its 1D analogue.

We derive an explicit expression for $\hat{Q}_t^{z,w}$ in Appendix E. The key takeaway is that $\hat{Q}_t^{z,w}$ not only depends on $p^{z,w}, \mathcal{Z}$, but also on new coefficients $\{c_x, d_x\}_{1 \le x \le N}$ that encode the steering effect of

guidance on marginals, which are defined as follows: for all $x_1, x_2 = 1, 2, \cdots, N-1$,

$$c_{x_1} := \frac{\sum_l p(x_1, l)^{-w} p(x_1, l|z)^{1+w}}{p(x_1)^{-w} p(x_1|z)^{1+w}}, \quad d_{x_2} = \frac{\sum_l p(l, x_2)^{-w} p(l, x_2|z)^{1+w}}{p(x_2)^{-w} p(x_2|z)^{1+w}}, \tag{12}$$

$$c_N := \frac{\sum_{l_1, l_2} p(l_1, l_2)^{-w} p(l_1, l_2|z)^{1+w}}{\sum_{l_1} p(l_1)^{-w} p(l_1|z)^{1+w}}, \quad d_N := \frac{\sum_{l_1, l_2} p(l_1, l_2)^{-w} p(l_1, l_2|z)^{1+w}}{\sum_{l_2} p(l_2)^{-w} p(l_2|z)^{1+w}}. \tag{13}$$

**Theorem 3.2** (2D reverse dynamics). *If $D = 2$ and $q_t^{z,w}$ satisfies the sampling dynamics equation 10, for all $0 \le t \le T$,*

$$q_t^{z,w}(x) = \begin{cases} \alpha_t(x) \mathcal{Z} p^{z,w}(x), & x_1, x_2 \ne N, \\ \alpha_t(x) \mathcal{Z} p^{z,w}(x_i), & x_i = N \ne x_{\setminus i}, \\ \alpha_t(x), & x_1 = x_2 = N. \end{cases} \tag{14}$$

*where $\mathcal{Z} = \sum_{x \in S} p(x)^{-w} p(x|z)^{1+w}$ and*

$$\alpha_t(x) = \begin{cases} -\frac{\frac{1}{c_{x_1}} + \frac{1}{d_{x_2}}}{\lambda_{N,N}^{z,w}} + \frac{\frac{r(x)^{c_{x_1}}}{c_{x_1}} + \frac{r(x)^{-\lambda_{N,N}^{z,w}}}{\lambda_{N,N}^{z,w}}}{\lambda_{N,N}^{z,w} + c_{x_1}} + \frac{\frac{r(x)^{d_{x_2}}}{d_{x_2}} + \frac{r(x)^{-\lambda_{N,N}^{z,w}}}{\lambda_{N,N}^{z,w}}}{\lambda_{N,N}^{z,w} + d_{x_2}}, & x_1, x_2 \ne N, \\ -\frac{1}{c_{x_1}(\lambda_{NN}^{z,w} + c_{x_1})} \left( r(t)^{c_{x_1}} - r(t)^{-\lambda_{NN}^{z,w}} \right), & x_1 \ne N = x_2, \\ -\frac{1}{d_{x_2}(\lambda_{NN}^{z,w} + d_{x_2})} \left( r(t)^{d_{x_2}} - r(t)^{-\lambda_{NN}^{z,w}} \right), & x_2 \ne N = x_1, \\ r(t)^{-\lambda_{NN}^{z,w}}, & x_1 = x_2 = N, \end{cases}$$

*with $\lambda_{NN}^{z,w} := -\mathcal{Z}(1/c_N + 1/d_N)$ and $r(t) := \frac{1 - e^{-(T-t)}}{1 - e^{-T}}$.*

Although the reverse dynamics looks complicated, its implications are striking.

**Remark 3.3** (Explicit expression for the generated distribution). *By evaluating $q_t^{z,w}$ at the final time, we find the generated distribution differs from the tilted distribution. Instead,*

$$q_T^{z,w}(x) = \frac{1/c_{x_1} + 1/d_{x_2}}{1/c_N + 1/d_N} p^{z,w}(x), \quad \forall x \in \{1, 2, \cdots, N-1\}^2. \tag{15}$$

**Proposition 3.3.** *Under Assumption 1.1, depending on the projected supports $\{\mathcal{X}_{k,d}\}_{1 \le k \le M, 1 \le d \le 2}$, the sampled distribution $q_T^{z_1,w}$ admits the following different behaviors:*

(1) *If $\mathcal{X}_{1,d} \cap \mathcal{X}_{k,d} = \emptyset$ for $d = 1, 2$ and all $k \ge 2$, we have $q_T^{z_1,w} = p(\cdot|z_1)$.*

(2) *If $S_{1,d} := \mathcal{X}_{1,d} \cap \left( \cup_{k=2}^M \mathcal{X}_{k,d} \right) \ne \emptyset$ for some $d = 1, 2$.*

$$q_T^{z_1,w}(x) \propto \begin{cases} A_1^{z_1,w} p(x|z_1), & x \in \mathcal{R}_1, \\ A_{2,i}^{z_1,w} p(x|z_1), & x \in \mathcal{R}_{2,i}, i = 1, 2, \\ A_3^{z_1,w} p(x|z_1), & x \in \mathcal{R}_3, \\ A_4^{z_1,w} p(x|z_1), & x \in \mathcal{R}_4, \\ 0, & otherwise. \end{cases}$$

*where $\mathcal{R}_1, \mathcal{R}_{2,1}, \mathcal{R}_{2,2}, \mathcal{R}_3, \mathcal{R}_4$ forms a partition of $\mathcal{X}_1$ and $A_1^{z_1,w}, A_{2,1}^{z_1,w}, A_{2,2}^{z_1,w}, A_3^{z_1,w}, A_4^{z_1,w}$ are associated weights on the regions. Explicit expression are given below:*

$$A_1^{z_1,w} = 2, \qquad \mathcal{R}_1 := \{x | x \in \mathcal{X}_1, x_1 \in \mathcal{X}_{1,1} \setminus S_{1,1}, x_2 \in \mathcal{X}_{1,2} \setminus S_{1,2}\},$$

$$A_{2,i}^{z_1,w} = 1 + \left( \frac{a_1 p(x_i|z_1)}{\sum_{k \in I_{1,i}} a_k p(x_i|z_k)} \right)^w, \quad \mathcal{R}_{2,i} := \{x | x \in \mathcal{X}_1, x_i \in S_{1,i}, x_{\setminus i} \in \mathcal{X}_{1,\setminus i} \setminus S_{1,\setminus i}\},$$

$$A_3^{z_1,w} = \sum_{i=1}^2 \left( \frac{a_1 p(x_i|z_1)}{\sum_{k \in I_{1,i}} a_k p(x_i|z_k)} \right)^w, \quad \mathcal{R}_3 := \{x | x \in \mathcal{X}_1 \setminus S_1, x_1 \in S_{1,1}, x_2 \in S_{1,2}\},$$

$$A_4^{z_1,w} = \left( \sum_{i=1}^2 \left( \frac{a_1 p(x_i|z_1)}{\sum_{k \in I_{1,i}} a_k p(x_i|z_k)} \right)^w \right) \left( \frac{a_1 p(x|z_1)}{\sum_{k \in I_1} a_k p(x|z_k)} \right)^w, \quad \mathcal{R}_4 := S_1.$$

*where $S_1 := \mathcal{X}_1 \cap \left( \cup_{k=2}^M \mathcal{X}_k \right), I_1 := \{k : \mathcal{X}_k \cap \mathcal{X}_1 \ne \emptyset\}, I_{1,d} := \{k : \mathcal{X}_{k,d} \cap \mathcal{X}_{1,d} \ne \emptyset\}.$*

*Last, as $w \to \infty$, $q_T^{z_1,w} \to q^{z_1,\infty}(\cdot|z_1)$ pointwisely, where $q^{z_1,\infty}(\cdot|z_1)$ satisfies that $\text{Supp}(q^{z_1,\infty}(\cdot|z_1)) \subset \mathcal{X}_1 \setminus S_1.$*

**Remark 3.4** (Effect of guidance on generated distributions). *In Proposition 3.3-(2), the generated distribution is a weighted version of class-1 distribution. The regions $\mathcal{R}_1, \mathcal{R}_{2,1}, \mathcal{R}_{2,2}, \mathcal{R}_3, \mathcal{R}_4$ reflect different level of "privacy" of class $z_1$: $\mathcal{R}_4$ is the overlapping region with other classes. $\mathcal{R}_1, \mathcal{R}_{2,i}, \mathcal{R}_3$ are not overlapping with other classes. But $\mathcal{R}_3$ has both projections overlapping with other classes and $\mathcal{R}_{2,i}$ has projections along dimension-$i$ overlapping with other classes. $\mathcal{R}_1$ is the most private set in class $z_1$, with no intersection with other classes even for projections. For all $w \geq 0$, the associated weights (before normalization) on different regions satisfies, $A_1^{z_1,w} \geq A_{2,i}^{z_1,2} \geq A_3^{z_1,w} \geq A_4^{z_1,w}$. This reflects that the sampled distribution from the discrete diffusion with CFG can leverage the geometric information of the full data distribution: the sampled distribution puts larger weights on more private regions of class $z_1$.*

To make the above discussion more concrete, Figure 1 illustrates a toy example where the full data distribution is a mixture of two uniform distributions supported on overlapping squares. We focus on sampling from the bottom-left class. The figure shows how classifier-free guidance (CFG) reshapes the generated distribution compared with the class-conditional distribution, redistributing probability mass away from overlapping regions as the guidance strength increases.

**Remark 3.5** (Discussion on $q^{z_1,\infty}(\cdot|z_1)$). *Under Assumption 1.1, $q^{z_1,\infty}(\cdot|z_1)$ has zero mass on overlapping region between class $z_1$ and other classes. In the non-overlapping region, explicit formula for $q^{z_1,\infty}(\cdot|z_1)$ can be derived from the expression of $q^{z_1,w}(\cdot|z_1)$ in Proposition 3.3. However, it strongly depends on the nullities of the regions in Proposition 3.3-(2). We refer the readers to Appendix E.1 for a detailed discussion.*

**Proposition 3.4** (2D convergence rate). *Under the conditions in Theorem 3.2, for all $0 \leq t \leq T$ and $w \gg 1$, $-\ln(\text{TV}(q_t^{z,w}, q_T^{z,w})) = \exp(\Theta(w)) \ln\left(\frac{1-e^{-T}}{1-e^{-(T-t)}}\right)$.*

As in 1D setting, the decay of TV in 2D exhibits a double-exponential dependency in $w$ as well.

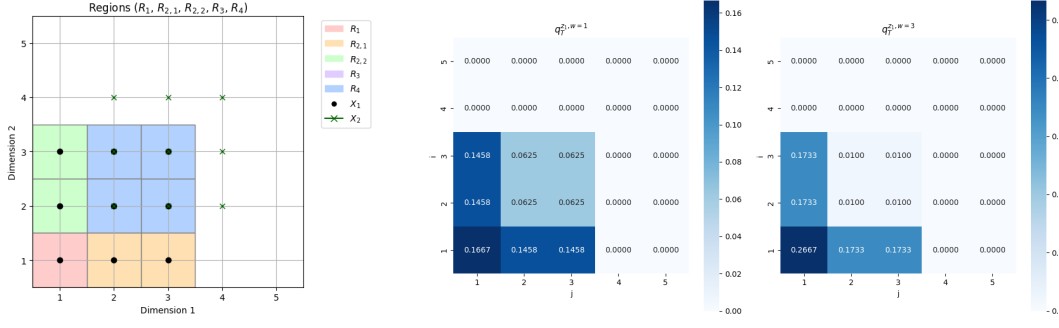

(a) Partition of the support into regions $\mathcal{R}_1, \mathcal{R}_{2,1}, \mathcal{R}_{2,2}, \mathcal{R}_3, \mathcal{R}_4$.

(b) Heatmaps: generated distribution $q_T^{z_1,w}$ for $w = 1$ (left), and $q_T^{z_1,w}$ for $w = 3$ (right).

Figure 1: Illustration of CFG effects in 2D: (a) how guided distributions change with $w$, (b) how class support is decomposed into regions reflecting different levels of overlap.

### 3.3 DISCUSSION: TOWARDS HIGHER DIMENSIONS

The explicit 1D and 2D analyses highlight two general principles of CFG in discrete diffusion. First, guidance reshapes the generated distribution by reallocating probability mass away from overlapping regions of class supports and toward class-specific regions. Second, the rate of convergence to the generated distribution exhibits a double-exponential dependence on the guidance parameter $w$. These effects are transparent in 1D and 2D, but the structure they reveal is suggestive of higher-dimensional behavior.

In higher dimensions ($D \geq 3$), we expect the interaction between classes supports to be governed not only by their full overlaps but also by overlaps of their marginal projections. More precisely:

- Unique regions: states in $\mathcal{X}_1$ whose projections are disjoint from the corresponding projections of all other classes should retain largest relative weight under CFG, with local mean/ covariance preserved.

- Partially overlapping regions: states that do not overlap with other classes but share marginal projections (e.g. along a subset of coordinates) are expected to be downweighted.

The strength of this downweighting should depend on ratios of conditional marginals, generalizing the coefficients $\{c_x, d_x\}$ from the 2D case.

- Fully overlapping regions: states in the intersection $\mathcal{X}_1 \cap \cup_{k=2}^M \mathcal{X}_k$ are suppressed as $w \to \infty$, with the probability mass redistributed towards the two types of regions above.

The convergence rate is also expected to follow the same pattern: the decay of total variation distance is exponential in time with an effective rate parameter that grows exponentially in $w$. In higher dimensions, although the rate will involve multiple interacting coefficients (generalizations of $\{c_x, d_x\}$) that encode how strongly the guidance penalizes overlap along different subsets of coordinates, the double-exponential dependency in $w$ is preserved due to the $\exp(w)$-dependency in the trace of the reverse rate matrix.

Overall, the 1D and 2D cases provide a blueprint: CFG can be understood as leveraging the geometry of overlaps in the support of the data distribution. In practice, this means that CFG tends to emphasize "private" regions of the target class and suppress ambiguous states, while accelerating convergence in a way that becomes sharper as $w$ increases. Extending these insights to formal results in higher dimensions remains an open problem, but our analysis offers a clear starting point for characterizing how guided discrete diffusion behaves in general settings.

## 4 NUMERICAL EXAMPLES

We now present numerical experiments that illustrate our theoretical results and probe their validity in higher dimensions. Unless otherwise stated, we use a small transformer to train the score model, Tau-leaping with 50 steps and log-linear schedule as the numerical scheme and 10K samples per experiment. All experiments are run on an NVIDIA GeForce RTX™ 4070 Laptop GPU.

**Experiments in** 1**D.** We examine two scenarios: when class supports are disjoint and when they overlap. From Figure 2-(a)(b), we can tell that: in the disjoint case, guidance has no effect. In contrast, when supports overlap, guidance redistributes mass away from the intersection region, in line with Proposition 3.1. Even with score and discretization errors, the sampled distribution closely approximates the tilted distribution. We also measure TV as a function of $w$ at a fixed time $t = .5$ in Figure 2-(c). For small $w$, the empirical TV curve matches our theoretical predictions. For large $w$, we observe a flat/increasing region in the plot. We conjecture that this is mainly due to the sharp transition of the reverse sampling dynamics for large $w$ (as shown in Remark 3.2), which makes the Tau-leaping scheme less efficient and less stable.

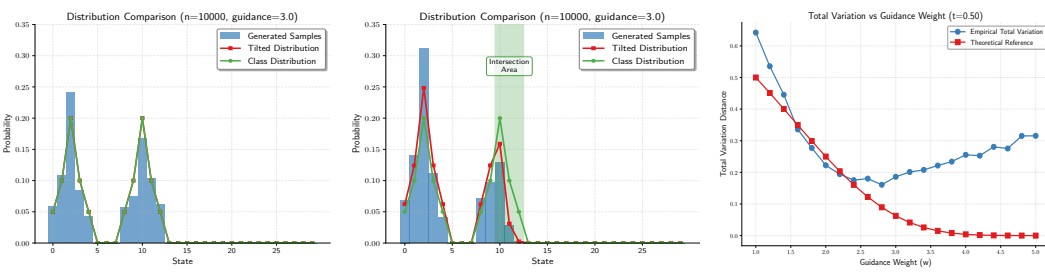

(a) No effect for disjoint support.  (b) Mass-shift when supports overlap.  (c) TV as a function of $w$.

Figure 2: The first two panels compare the generated distribution $q_T^{z,w}$ (blue histograms), the tilted distribution $p^{z,w}$ (red), and the true target class distribution $p(\cdot|z)$ (green). In (a), when the supports are disjoint, all three distributions align closely, indicating that guidance has no effect. In (b), when supports overlap, $q_T^{z,w}$ and $p^{z,w}$ remain close—consistent with Proposition 3.1—but both deviate from $p(\cdot|z)$, reflecting how CFG suppresses mass in overlapping regions and amplifies it in class-unique regions. Panel (c) plots the TV distance as a function of $w$ (blue), showing good agreement with the theoretical prediction from Proposition 3.2 (red) for small $w$.

**Experiments in** 2**D.** Figure 2 compares theoretical predictions with empirical results in 2D. Each class consists of two diamond-shaped regions: one at a corner and one centered (see Appendix F.2 for density plots). For the class of interest, the corner diamond lies in the bottom-left, while for the competing class it lies in the top-right. We consider two setups depending on whether the central

diamonds overlap. In the disjoint case, the classes do not intersect in the full space, but their projections overlap along coordinate axes, causing guidance to suppress probability mass at the top and right corners of the central diamond. In the overlapping case, the effect is more pronounced: the upper-right region of the central diamond, where supports coincide, is strongly suppressed, while projected overlaps slightly reduce mass in the upper-left and bottom-right regions. In both cases, the empirical results align closely with our theoretical predictions, validating of our analysis in 2D.

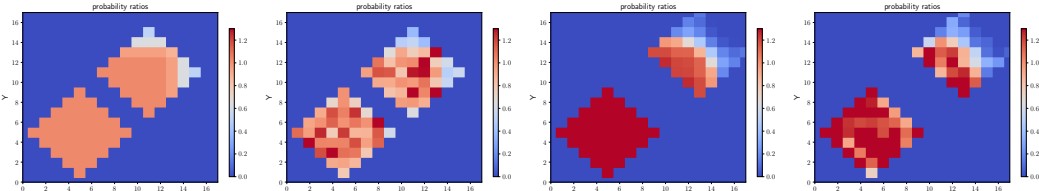

(a) Disjoint: ratio of predicted to class distribution. (b) Disjoint: ratio of generated to class distribution. (c) Overlap: ratio of predicted to class distribution. (d) Overlap: ratio of generated to class distribution.

Figure 3: Ratios of the generated distribution $q_T^{z,w}$ to the class distribution $p(\cdot|z)$ in 2D. Red indicates regions where mass is amplified, blue indicates suppression. (a,c) theoretical predictions from Proposition 3.3; (b,d) empirical results using Tau-leaping. In the disjoint case (a,b), guidance suppresses top and right corners of the central diamond due to overlap in projected spaces. In the overlapping case (c,d), guidance strongly suppresses the upper-right part of the central diamond and more moderately reduces mass in the upper-left and bottom-right parts, consistent with theory.

**Higher-dimensional Experiments.** To investigate whether our findings extend beyond low dimensions, we conduct experiments in 5D using mixtures of uniform hypercubes, $\{0, 1, 2\}^5$ and $\{2, 3, 4\}^5$, which overlap only at the single state $(2, 2, 2, 2, 2)^\top$. We focus on sampling from the class supported on $\{2, 3, 4\}^5$ under guidance strengths $w = 1$ and $w = 2$. The geometry naturally partitions the space by the number of coordinates equal to 2, denoted by $k = \#\{d : x_d = 2\}$. The case $k = 0$ corresponds to states unique to the target class, $k = 5$ corresponds to the fully overlapping state, and intermediate values $k \in \{1, 2, 3, 4\}$ represent partially overlapping regions.

Tables 1 and 2 report the number of states in each region along with the mean and standard deviation of their probabilities from empirical samples. In both cases we see a clear monotone trend: per-state mass is largest in the unique region ($k = 0$), smallest in the fully overlapping region ($k = 5$), and intermediate for partial overlaps. Importantly, partial-overlap regions are not uniformly suppressed; depending on $k$, they may either gain or lose mass relative to $p(\cdot|z)$.

Comparing $w = 1$ and $w = 2$, the redistribution becomes more pronounced with stronger guidance. Larger $w$ shifts probability mass away from the fully overlapping and high-overlap regions, reallocating it toward unique states and those with smaller $k$. These results mirror the mechanisms observed in 2D, providing strong evidence that our theoretical insights extend to higher dimensions.

Table 1: Sample statistics in regions defined by $\#\{d : x_d = 2\}$ for $w = 1$

| $\#\{d : x_d = 2\}$ | 0 | 1 | 2 | 3 | 4 | 5 |
|---|---|---|---|---|---|---|
| state frequency | 32 | 80 | 80 | 40 | 10 | 1 |
| density per state - average (1e-3) | 4.984 | 4.541 | 4.020 | 3.386 | 2.208 | 0.280 |
| density per state - std (1e-3) | 0.282 | 0.319 | 0.268 | 0.273 | 0.206 | 0.000 |

Table 2: Sample statistics in regions defined by $\#\{d : x_d = 2\}$ for $w = 2$

| $\#\{d : x_d = 2\}$ | 0 | 1 | 2 | 3 | 4 | 5 |
|---|---|---|---|---|---|---|
| state frequency | 32 | 80 | 80 | 40 | 10 | 1 |
| density per state - average (1e-3) | 5.573 | 4.783 | 3.867 | 2.844 | 1.588 | 0.200 |
| density per state - std (1e-3) | 0.346 | 0.312 | 0.268 | 0.204 | 0.171 | 0.000 |

**Additional Experiments.** We further explore the effect of CFG in masked discrete generation on MNIST. Full details of it, together with details of previous experiments, are provided in Appendix F.

## 5 CONCLUSION, LIMITATION AND FUTURE WORK

In this paper, we developed a rigorous framework for understanding classifier-free guidance (CFG) in masked discrete diffusion models. Through explicit analysis in one and two dimensions, we showed how guidance reshapes generated distributions: in 1D the guided process exactly recovers the tilted distribution, while in 2D systematic deviations emerge that can be described in terms of class overlaps and marginal reweighting. These tractable cases shed light on what to expect in higher dimensions, where overlaps and geometry are more complex. A key theoretical finding is that the total variation distance along the reverse dynamics decays double-exponentially with the guidance strength $w$. While this implies strong contraction in the continuous-time limit, it also explains why large $w$ leads to stiff and unstable numerical simulation in practice.

**Limitation.** Our analysis is limited to low-dimensional settings (1D and 2D). Extending to higher dimensions is challenging for two reasons: exact formulas are unlikely to exist, as the reverse rate matrix becomes prohibitively complex, and even approximate descriptions that capture the qualitative role of guidance are difficult to derive due to intricate interactions between overlapping supports and marginal projections. These challenges restrict the scope of our current theoretical guarantees.

**Future Work.** A natural next step is to relax the idealized assumptions of exact concrete scores and perfect numerical integration. Important open questions include how guidance interacts with score approximation errors and discretization errors, whether it amplifies or mitigates them, and how such interactions affect convergence and sample quality. Progress on these fronts would deepen the theoretical foundations of guided discrete diffusion and improve its reliability in practice.

## ACKNOWLEDGMENTS

YH, KR and MT are thankful for partial supports by NSF Grants DMS-1847802, DMS-2513699, DOE Grants NA0004261, SC0026274, and Richard Duke Fellowship.

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

CONTENTS

## A ADDITIONAL RELATED WORK

**Diffusion Models in Continuous and Discrete Spaces.** Diffusion models were first developed in continuous domains, where Gaussian noise is gradually added and then removed through a learned reverse process (Ho et al., 2020; Song et al., 2021). They have achieved remarkable success for images (Dhariwal & Nichol, 2021) and audio (Kong et al., 2021), but are less natural for inherently discrete data such as text or categorical variables. To address this, discrete diffusion models have been proposed, including D3PMs (Austin et al., 2021) and masked token diffusion models (Campbell et al., 2022; Shi et al., 2024; Ou et al., 2024), which corrupt data via masking or categorical transitions. Compared with continuous models, discrete diffusion introduces new challenges: forward marginals are typically non-Gaussian and analytically intractable, while score functions must be redefined, often as ratios of logits (Campbell et al., 2022; Lou et al., 2023). Masked diffusion alleviates some of these issues, since its corruption process admits closed-form marginals and enables tractable likelihood training (Shi et al., 2024; Ou et al., 2024), making it both practical and amenable to analysis.

**Guidance in Continuous Diffusion Models.** Guidance techniques are central to controllable generation in continuous diffusion. Classifier guidance (Dhariwal & Nichol, 2021) steers the reverse dynamics using gradients from a pretrained classifier, but requires large and accurate auxiliary models. Classifier-free guidance (CFG) (Ho & Salimans, 2021) eliminates this dependence by jointly training unconditional and conditional models, allowing interpolation between guided and unguided generations through a scaling parameter. Recent theoretical work has begun to analyze these mechanisms. Assuming Gaussian structure, Bradley & Nakkiran (2024) showed that the probability flow ODE under guidance does not recover the tilted distribution, and is equivalent to a predictor–corrector scheme. Wu et al. (2024) studied Gaussian mixtures, proving that increasing guidance reduces differential entropy and sharpens class confidence. Chidambaram et al. (2024) analyzed guided ODE dynamics in 1D mixtures, showing that guidance exploits latent geometric information even when absent from the classifier. While insightful, these results remain restricted to highly simplified settings.

**Guidance in Discrete Diffusion Models.** For discrete data, several approaches apply guidance indirectly through continuous embeddings (Li et al., 2022; Han et al., 2022; Lovelace et al., 2023; Stark et al., 2024; Guo et al., 2024). More recently, direct formulations have emerged: Nisonoff et al. (2024) proposed modifying reverse transition rate matrices, while Schiff et al. (2024) guided the reverse kernels. These definitions need not coincide, and systematic comparisons remain open. Our work focuses on the framework of Nisonoff et al. (2024).

To our knowledge, this is the first theoretical study of guidance in discrete diffusion models. Closely related to Chidambaram et al. (2024) in spirit, we analyze low-dimensional sampling dynamics where explicit solutions are available. Our results show that in 1D, guided sampling exactly recovers the tilted distribution, while in 2D discrepancies arise that can be characterized analytically. Moreover, we prove that increasing guidance suppresses probability in overlapping regions and amplifies unique regions, reducing diversity as guidance grows. The total variation distance along the reverse process decays double-exponentially in the guidance parameter, highlighting both the sharp concentration effects and potential numerical instabilities at large guidance values. These findings parallel—but also extend—observations in continuous settings (Bradley & Nakkiran, 2024; Wu et al., 2024; Chidambaram et al., 2024), providing new insight into the discrete domain.

## B ANALYSIS OF MASKED DISCRETE DIFFUSION MODELS WITHOUT GUIDANCE

This section analyzes the behavior of masked discrete diffusion in the absence of guidance. By quantifying the density evolution of the sampling process, we establish a baseline understanding of the unguided dynamics. These results provide essential groundwork for the theoretical analysis of discrete diffusion with CFG in the Section 3.

## B.1 DENSITY EVOLUTION ALONG THE FORWARD PROCESS

The forward process in the masked discrete diffusion process gradually absorbs all the mass to the masked state $N$. In practice (Campbell et al., 2022; Lou et al., 2023), the forward transition rate matrix is parametrized by $Q_t = \sigma(t)\big(\sum_{d=1}^{D} I_N \otimes \cdots \underbrace{Q}_{d^{th}} \cdots \otimes I_N\big)$ where

$$Q = \begin{pmatrix} -1 & \cdots & 0 & 0 \\ \vdots & \ddots & \vdots & \vdots \\ 0 & \cdots & -1 & 0 \\ 1 & \cdots & 1 & 0 \end{pmatrix}_{N \times N} \tag{16}$$

For simplicity, we consider $\sigma(t) \equiv 1$ in the paper. According equation 16, we express the densities along the forward process in the following proposition whose proof is deferred to Appendix C.

**Proposition B.1.** *Let $\mu_t$ be the solution to $\frac{\mathrm{d}}{\mathrm{d}t}\mu_t = Q_t \mu_t$ with initial distribution $\mu_0 = \mu$ and $Q_t$ given above. Then*

$$\mu_t = \begin{pmatrix} e^{-t} & 0 & \cdots & 0 & 0 \\ 0 & e^{-t} & \cdots & 0 & 0 \\ \vdots & \vdots & \ddots & \vdots & \vdots \\ 0 & 0 & \cdots & e^{-t} & 0 \\ 1-e^{-t} & 1-e^{-t} & \cdots & 1-e^{-t} & 1 \end{pmatrix}_{N \times N}^{\otimes D} \mu := A_t^{\otimes D}\mu. \tag{17}$$

*As a consequence, for any $x \in S$ with $\mathrm{UM} = \mathrm{UM}(x) := |\{i : x_i < N\}|$,*

$$\mu_t(x) = e^{-|\mathrm{UM}|t}(1 - e^{-t})^{D - |\mathrm{UM}|} \sum_{y:y_{\mathrm{UM}}=x_{\mathrm{UM}}} \mu(y).$$

## B.2 DENSITY EVOLUTION ALONG THE REVERSE PROCESS

For any distribution $\mu$ on $S$, the forward process equation 1 initiated at $\mu$ induces a reverse process, whose transition rate matrix is denoted as $\bar{Q}_t[\mu]$. An explicit expression of $\bar{Q}_t[\mu]$ can be derived from the property in equation 3 and the forward densities in Proposition B.1.

**Proposition B.2.** *The sequence of reverse transition rate matrices associated with $\mu$ (initialization of the forward process) satisfies that for all $0 \le t \le T$,*

$$\bar{Q}_t[\mu](y,x) = \begin{cases} \frac{e^{-t}}{1-e^{-t}} \frac{\sum_{u:u_{\mathrm{UM}}=y_{\mathrm{UM}}} \mu(u)}{\sum_{u:u_{\mathrm{UM}}=x_{\mathrm{UM}}} \mu(u)}, & x_i \ne N = y_i, x_{\backslash i} = y_{\backslash i}, \\ -\sum_{u \in \mathcal{N}(x)} \bar{Q}_t[\mu](u,x), & y = x, \\ 0, & \text{otherwise.} \end{cases}$$

With the above expression of the reverse transition rate matrix, in the low-dimensional setting of $D = 1$, we derive the density formulas along the reverse sampling dynamics in the following theorem, with the proof deferred to Appendix C.

**Theorem B.1.** *In the discrete diffusion models on $S$, if $D = 1$ and $q_t$ satisfies the sampling dynamics $\frac{\mathrm{d}}{\mathrm{d}t}q_t = \bar{Q}_{T-t}[p]q_t$ with initial condition $q_0 = \delta_N$, we have that for all $0 \le t \le T$,*

$$q_t(x) = \begin{cases} \left(1 - \frac{1-e^{-(T-t)}}{1-e^{-T}}\right)p(x), & x = 1, 2, \cdots, N-1, \\ \frac{1-e^{-(T-t)}}{1-e^{-T}}, & x = N. \end{cases} \tag{18}$$

**Remark B.1** (No initialization error). *Unlike other diffusion processes, the absorbing discrete diffusion does not induce any initialization error. Even though we approximate the initialization in equation 2 by the point mass at the masked state, the sampled distribution recovers the data distribution, i.e., $q_T = p$. Same property also holds for masked discrete diffusion with CFG, as shown in Section 3.*

## C PROOFS - PROPERTIES OF MASKED DISCRETE DIFFUSION MODELS WITHOUT GUIDANCE

**Lemma C.1** (Diagonalization of $Q$). $Q = X\Lambda X^{-1}$ with $\Lambda = \text{Diag}(-1,\cdots,-1,0)$ and

$$X = X^{-1} = \begin{pmatrix} -1 & -1 & \cdots & -1 & -1 & 0 \\ 1 & 0 & \cdots & 0 & 0 & 0 \\ 0 & 1 & \cdots & 0 & 0 & 0 \\ \vdots & \vdots & \ddots & \vdots & \vdots & \vdots \\ 0 & 0 & \cdots & 1 & 0 & 0 \\ 0 & 0 & \cdots & 0 & 1 & 1 \end{pmatrix}_{N\times N}$$

*Proof of Proposition B.1.* The solution to equation 1 with initial distribution $\mu_0 = \mu$ can be expressed as

$$\mu_t = \exp(tQ)\mu = \exp\left(t\sum_{d=1}^{D} I_N \otimes \cdots \underbrace{Q}_{d^{th}} \cdots \otimes I_N\right)\mu$$

$$= \prod_{d=1}^{D} \exp\left(tI_N \otimes \cdots \underbrace{Q}_{d^{th}} \cdots \otimes I_N\right)\mu$$

$$= \prod_{d=1}^{D} \exp\left(t(X \otimes \cdots \otimes X)\left(I_N \otimes \cdots \underbrace{\Lambda}_{d^{th}} \cdots \otimes I_N\right)(X \otimes \cdots \otimes X)^{-1}\right)\mu$$

$$= \prod_{d=1}^{D}(X \otimes \cdots \otimes X)\exp\left(I_N \otimes \cdots \underbrace{t\Lambda}_{d^{th}} \cdots \otimes I_N\right)(X \otimes \cdots \otimes X)^{-1}\mu$$

$$= (X \otimes \cdots \otimes X)\exp(t\Lambda)^{\otimes D}(X \otimes \cdots \otimes X)^{-1}\mu$$

$$= \left(X\exp(t\Lambda)X^{-1}\right)^{\otimes D}\mu,$$

where the second identity uses the fact that $(I_N \otimes \cdots \underbrace{Q}_{d^{th}} \cdots \otimes I_N)_d$ commute with each other.

Then the statement follows from Lemma C.1. □

*Proof of Theorem B.1.* With the expression of the distribution along the forward process, we can write the reverse transition rate matrix based on Proposition B.2. We have

$$\bar{Q}_t = \frac{e^{-t}}{1-e^{-t}}\bar{Q} := \frac{e^{-t}}{1-e^{-t}}\begin{pmatrix} 0 & 0 & \cdots & 0 & p(1) \\ 0 & 0 & \cdots & 0 & p(2) \\ \vdots & \vdots & \ddots & \vdots & \vdots \\ 0 & 0 & \cdots & 0 & p(N-1) \\ 0 & 0 & \cdots & 0 & -1 \end{pmatrix}_{N\times N} \tag{19}$$

The eigenvalues and eigenvectors of $\bar{Q}$ are given by

$$\bar{\lambda}_1 = \bar{\lambda}_2 = \cdots = \bar{\lambda}_{N-1} = 0, \quad \bar{\lambda}_N = -1,$$

$$\vec{u}_1 = \begin{pmatrix} 1 \\ 0 \\ \vdots \\ 0 \\ 0 \end{pmatrix}, \vec{u}_2 = \begin{pmatrix} 0 \\ 1 \\ \vdots \\ 0 \\ 0 \end{pmatrix}, \cdots, \vec{u}_{N-1} = \begin{pmatrix} 0 \\ 0 \\ \vdots \\ 1 \\ 0 \end{pmatrix}, \vec{u}_N = \begin{pmatrix} p(1) \\ p(2) \\ \vdots \\ p(N-1) \\ -1 \end{pmatrix}$$

The eigenvalue decomposition of $\bar{Q}$ is given by $\bar{Q} = \bar{X}\bar{D}\bar{X}^{-1}$ with $\bar{D} = \mathrm{diag}(0, 0, \cdots, 0, -1) \in \mathbb{R}^{N \times N}$

$$\bar{X} = \bar{X}^{-1} = \begin{pmatrix} 1 & 0 & \cdots & 0 & p(1) \\ 0 & 1 & \cdots & 0 & p(2) \\ \vdots & \vdots & \ddots & \vdots & \vdots \\ 0 & 0 & \cdots & 1 & p(N-1) \\ 0 & 0 & \cdots & 0 & -1 \end{pmatrix}_{N \times N}.$$

A simple computation tells that

$$\exp\Big(\int_0^{T-t} \bar{Q}_{T-s}\mathrm{d}s\Big) = \exp\Big(\int_t^T \frac{e^{-s}}{1-e^{-s}}\mathrm{d}s\bar{Q}\Big) = \bar{X}\exp\Big(\ln(\frac{1-e^{-T}}{1-e^{-t}})\bar{D}\Big)\bar{X}^{-1}$$

$$= \begin{pmatrix} 1 & 0 & \cdots & 0 & \big(1-\frac{1-e^{-t}}{1-e^{-T}}\big)p(1) \\ 0 & 1 & \cdots & 0 & \big(1-\frac{1-e^{-t}}{1-e^{-T}}\big)p(2) \\ \vdots & \vdots & \ddots & \vdots & \vdots \\ 0 & 0 & \cdots & 1 & \big(1-\frac{1-e^{-t}}{1-e^{-T}}\big)p(N-1) \\ 0 & 0 & \cdots & 0 & \frac{1-e^{-t}}{1-e^{-T}} \end{pmatrix}_{N \times N}.$$

Along the reverse sampling dynamics, we have $q_t = \exp\big(\int_0^t \bar{Q}_{T-s}\mathrm{d}s\big)q_0$, which implies

$$q_t(x) = \begin{cases} q_0(x) + \big(1 - \frac{1-e^{-(T-t)}}{1-e^{-T}}\big)p(x)q_0(N), & x = 1, 2, \cdots, N-1, \\ \frac{1-e^{-(T-t)}}{1-e^{-T}}q_0(N), & x = N. \end{cases} \tag{20}$$

Last, the theorem follows from plugging in $q_0 = \delta_N$. $\qquad\square$

## D  PROPERTIES OF MASKED DISCRETE DIFFUSION MODELS WITH CFG WHEN $D = 1$

*Proof of Theorem 3.1.* Notice that the reverse transition rate matrix $\hat{Q}_t^{z,w} = \mathcal{Z}^{z,w}\hat{Q}_t[p^{z,w}] := \mathcal{Z}\hat{Q}_t$. Following the same computation in the proof of Theorem B.1, we have

$$\exp\Big(\int_0^{T-t}\mathcal{Z}\hat{Q}_{T-s}\mathrm{d}s\Big) = \exp\Big(\mathcal{Z}\int_t^T\frac{e^{-s}}{1-e^{-s}}\mathrm{d}s\hat{Q}\Big) = \bar{X}\exp\Big(\mathcal{Z}\ln(\frac{1-e^{-T}}{1-e^{-t}})\bar{D}\Big)\bar{X}^{-1}$$

$$= \begin{pmatrix} 1 & 0 & \cdots & 0 & \big(1-(\frac{1-e^{-t}}{1-e^{-T}})^{\mathcal{Z}}\big)p^{z,w}(1) \\ 0 & 1 & \cdots & 0 & \big(1-(\frac{1-e^{-t}}{1-e^{-T}})^{\mathcal{Z}}\big)p^{z,w}(2) \\ \vdots & \vdots & \ddots & \vdots & \vdots \\ 0 & 0 & \cdots & 1 & \big(1-(\frac{1-e^{-t}}{1-e^{-T}})^{\mathcal{Z}}\big)p^{z,w}(N-1) \\ 0 & 0 & \cdots & 0 & (\frac{1-e^{-t}}{1-e^{-T}})^{\mathcal{Z}} \end{pmatrix}_{N \times N}.$$

Along the reverse sampling dynamics equation 10, we have $q_t^{z,w} = \exp\big(\int_0^t \mathcal{Z}\hat{Q}_{T-s}\mathrm{d}s\big)p_T^{z,w}$, which implies

$$q_t^{z,w}(x) = \begin{cases} q_0^{z,w}(x) + \big(1 - \frac{1-e^{-(T-t)}}{1-e^{-T}}\big)^{\mathcal{Z}}p^{z,w}(x)q_0^{z,w}(N), & x = 1, 2, \cdots, N-1, \\ \big(\frac{1-e^{-(T-t)}}{1-e^{-T}}\big)^{\mathcal{Z}}q_0^{z,w}(N), & x = N. \end{cases} \tag{21}$$

Last, the theorem follows from plugging in $q_0^{z,w} = \delta_N$. $\qquad\square$

*Proof of Proposition 3.1.* According to Theorem 3.1, in both cases, the sampled distribution is the same as the tilted distribution, i.e., $q_T^{z_1,w} = p^{z_1,w}$.

In case (1), it is obvious that $p^{z_1,w} = p(\cdot|z_1)$.

In case (2), we have $p^{z_1,w}(x) \propto (\frac{p(x|z_1)}{p(x)})^w p(x|z_1)$. Under Assumption 1.1, we have

$$\frac{p(x|z_1)}{p(x)} = \begin{cases} \frac{p(x|z_1)}{a_1 p(x|z_1)}, & x \in \mathcal{X}_1 \setminus S_1 \\ \frac{p(x|z_1)}{\sum_k a_k p(x|z_k)}, & x \in S_1, \\ 0, & \text{otherwise.} \end{cases}$$

Then Proposition 3.1-(2) is proved. $\qquad\square$

*Proof of Proposition 3.2.* The result directly follows from Theorem 3.1 and the formula $\mathrm{TV}(\mu_1, \mu_2) = \frac{1}{2} \sum_x |\mu_1(x) - \mu_2(x)|$. $\qquad\square$

**Definition D.1** (Local mean and covariance)**.** *For any probability distribution $\mu$ on $S$ and any subset $A \subset S$, the local mean and local covariance of $\mu$ on $A$ are defined respectively as*

$$m_A(\mu) := \sum_{x \in A} x \mu_A(x), \quad \Sigma_A(\mu) := \sum_x (x - m_A(\mu))(x - m_A(x))^\intercal \mu_A(x),$$

*where $\mu_A(x) := \mu(x) / \sum_{y \in A} \mu(y)$ is the restriction of $\mu$ on $A$.*

**Lemma D.1.** *Under the assumptions in Proposition 3.1, for all $w > 0$, $\Sigma_{\mathcal{X}_1 \setminus S_1}(q_T^{z_1,w}) = \Sigma_{\mathcal{X}_1 \setminus S_1}(p(\cdot|z_1))$.*

*Proof of Lemma D.1.* According to Proposition 3.1,

$$m_{\mathcal{X}_1 \setminus S_1}(q_T^{z_1,w}) = \sum_{x \in \mathcal{X}_1 \setminus S_1} x p(x|z_1) / \sum_{y \in \mathcal{X}_1 \setminus S_1} p(y|z_1) = m_{\mathcal{X}_1 \setminus S_1}(p(\cdot|z_1)),$$

and

$$\begin{aligned} \Sigma_{\mathcal{X}_1 \setminus S_1}(q_T^{z_1,w}) &= \sum_{x \in \mathcal{X}_1 \setminus S_1} (x - m_{\mathcal{X}_1 \setminus S_1}(q_T^{z_1,w}))(x - m_{\mathcal{X}_1 \setminus S_1}(q_T^{z_1,w}))^\intercal p(x|z_1) / \sum_{y \in \mathcal{X}_1 \setminus S_1} p(y|z_1) \\ &= \sum_{x \in \mathcal{X}_1 \setminus S_1} (x - m_{\mathcal{X}_1 \setminus S_1}(p(\cdot|z_1)))(x - m_{\mathcal{X}_1 \setminus S_1}(p(\cdot|z_1))^\intercal p(x|z_1) / \sum_{y \in \mathcal{X}_1 \setminus S_1} p(y|z_1) \\ &= \Sigma_{\mathcal{X}_1 \setminus S_1}(p(\cdot|z_1)). \end{aligned}$$

$\qquad\square$

# E PROPERTIES OF MASKED DISCRETE DIFFUSION MODELS WITH CFG WHEN $D = 2$

**Proposition E.1.** *When $D = 2$, denote $\mathcal{Z} = \mathcal{Z}^{z,w} = \sum_{x \in S} p(x)^{-w} p(x|z)^{1+w}$. Then the guided reverse transition rate matrix is given by $\hat{Q}_t^{z,w} = \frac{e^{-t}}{1-e^{-t}} \hat{Q}^{z,w}$ s.t.,*

$$\hat{Q}^{z,w}(y,x) = \begin{cases} \frac{\mathcal{Z} p^{z,w}(y)}{p(y_i)^{-w} p(y_i|z)^{1+w}}, & x_i = y_i \neq N, x_{\setminus i} = N \neq y_{\setminus i} \\ p(y_i)^{-w} p(y_i|z)^{1+w}, & x_i = N \neq y_i, x_{\setminus i} = y_{\setminus i} = N \\ -\frac{\mathcal{Z} p^{z,w}(y_i)}{p(y_i)^{-w} p(y_i|z)^{1+w}}, & x_i = y_i \neq N, x_{\setminus i} = y_{\setminus i} = N \\ -\sum_{l=1}^2 \sum_{u_l=1}^{N-1} p(u_1)^{-w} p(u_l|z)^{1+w}, & x = y = (N, N) \\ 0, & \text{otherwise.} \end{cases}$$

*Proof of Proposition E.1.* For any $x, y \in S$ with $x_i = y_i \neq N$ and $x_j = N \neq y_j$, according to equation 9, we have

$$\begin{aligned} \hat{Q}_t^{z,w}(y,x) &= \bar{Q}_t^z(y,x)^{-w} \bar{Q}_t(y,x)^{1+w} = \left(\frac{p_t(y)}{p_t(x)}\right)^{-w} \left(\frac{p_t(y|z)}{p_t(x|z)}\right)^{1+w} \\ &= \left(\frac{e^{-2t} p(y)}{e^{-t}(1-e^{-t}) p(y_i)}\right)^{-w} \left(\frac{e^{-2t} p(y|z)}{e^{-t}(1-e^{-t}) p(y_i|z)}\right)^{1+w} \end{aligned}$$

$$= \frac{e^{-t}}{1 - e^{-t}} \frac{p(y)^{-w} p(y|z)^{1+w}}{p(y_i)^{-w} p(y_i|z)^{1+w}}$$

$$= \frac{e^{-t}}{1 - e^{-t}} \frac{\mathcal{Z}^{z,w} p^{z,w}(y)}{p(y_i)^{-w} p(y_i|z)^{1+w}},$$

where the third identity follows from Proposition B.1, and the last identity follows from the definition of $p^{z,w}$. Next, following the same approach, for any $x, y \in S$ with $x_i = N \neq y_i$ and $x_j = y_j = N$, we have

$$\hat{Q}_t^{z,w}(y, x) = \left(\frac{p_t(y)}{p_t(x)}\right)^{-w} \left(\frac{p_t(y|z)}{p_t(x|z)}\right)^{1+w}$$

$$= \left(\frac{e^{-t}(1 - e^{-t})p(y_i)}{(1 - e^{-t})^2}\right)^{-w} \left(\frac{e^{-t}(1 - e^{-t})p(y_i|z)}{(1 - e^{-t})^2}\right)^{1+w}$$

$$= \frac{e^{-t}}{1 - e^{-t}} p(y_i)^{-w} p(y_i|z)^{1+w}$$

Last, the other cases for different $(y, x)$ follows from the definition of the transition rate matrix,0 equation 3 and equation 8. □

*Proof of Theorem 3.2.* Our proof follows from the following steps.

Step 1: represent the reverse transition rate matrix blockwisely. The matrix $Q^{z,w}$ in Proposition E.1 can be represented blockwisely as

$$\hat{Q}^{z,w} = \begin{pmatrix} \hat{R}_1^{z,w} & \cdots & \mathbf{0} & \hat{L}_1^{z,w} \\ \vdots & \ddots & \vdots & \cdots \\ \mathbf{0} & \cdots & \hat{R}_{N-1}^{z,w} & \hat{L}_{N-1}^{z,w} \\ \mathbf{0} & \cdots & \mathbf{0} & \hat{M}^{z,w} - \sum_i \hat{L}_i^{z,w} \end{pmatrix},$$

For all $i = 1, 2, \cdots N - 1$,

$$\hat{R}_i^{z,w} := \begin{pmatrix} 0 & 0 & \cdots & \left(\frac{p(i,1)}{\sum_l p(i,l)}\right)^{-w} \left(\frac{p(i,1|z)}{\sum_l p(i,l|z)}\right)^{1+w} \\ 0 & 0 & \cdots & \left(\frac{p(i,2)}{\sum_l p(i,l)}\right)^{-w} \left(\frac{p(i,2|z)}{\sum_l p(i,l|z)}\right)^{1+w} \\ \vdots & \vdots & \ddots & \vdots \\ 0 & 0 & \cdots & -\sum_j \left(\frac{p(i,j)}{\sum_l p(i,l)}\right)^{-w} \left(\frac{p(i,j|z)}{\sum_l p(i,l|z)}\right)^{1+w} \end{pmatrix}, \tag{22}$$

$$\hat{L}_i^{z,w} := \tag{23}$$

$$\begin{pmatrix} \left(\frac{p(i,1)}{\sum_l p(l,1)}\right)^{-w} \left(\frac{p(i,1|z)}{\sum_l p(l,1|z)}\right)^{1+w} & \cdots & 0 & 0 \\ \vdots & \ddots & \vdots & 0 \\ 0 & \cdots & \left(\frac{p(i,N-1)}{\sum_l p(l,N-1)}\right)^{-w} \left(\frac{p(i,N-1|z)}{\sum_l p(l,N-1|z)}\right)^{1+w} & \vdots \\ 0 & 0 & \cdots & \left(\sum_l p(i,l)\right)^{-w} \left(\sum_l p(i,l|z)\right)^{1+w} \end{pmatrix},$$

$$\hat{M}^{z,w} := \begin{pmatrix} 0 & 0 & \cdots & \left(\sum_l p(l,1)\right)^{-w} \left(\sum_l p(l,1|z)\right)^{1+w} \\ 0 & 0 & \cdots & \left(\sum_l p(l,2)\right)^{-w} \left(\sum_l p(l,2|z)\right)^{1+w} \\ \vdots & \vdots & \ddots & \vdots \\ 0 & 0 & \cdots & -\sum_j \left(\sum_l p(l,j)\right)^{-w} \left(\sum_l p(l,j|z)\right)^{1+w} \end{pmatrix}, \tag{24}$$

where we used the definition of $\mathcal{Z}$ and marginal distributions.

Step 2: Eigenvalue decomposition for $\hat{Q}^{z,w}$ Since $\hat{Q}^{z,w}$ is upper triangular, its eigenvalues are diagonal entries. For all $i, j = 1, 2, \cdots, N - 1$, define

$$c_i := \frac{\sum_l p(i,l)^{-w} p(i,l|z)^{1+w}}{\left(\sum_l p(i,l)\right)^{-w} \left(\sum_l p(i,l|z)\right)^{1+w}}, \quad d_j := \frac{\sum_l p(l,j)^{-w} p(l,j|z)^{1+w}}{\left(\sum_l p(l,j)\right)^{-w} \left(\sum_l p(l,j|z)\right)^{1+w}}, \tag{25}$$

$$c_N := \frac{\sum_{l_1,l_2} p(l_1,l_2)^{-w} p(l_1,l_2|z)^{1+w}}{\sum_{l_1} \left(\sum_{l_2} p(l_1,l_2)\right)^{-w} \left(\sum_{l_2} p(l_1,l_2|z)\right)^{1+w}}, \quad d_N := \frac{\sum_{l_1,l_2} p(l_1,l_2)^{-w} p(l_1,l_2|z)^{1+w}}{\sum_{l_2} \left(\sum_{l_1} p(l_1,l_2)\right)^{-w} \left(\sum_{l_1} p(l_1,l_2|z)\right)^{1+w}}. \tag{26}$$

Then the set of eigenvalues for $\hat{Q}^{z,w}$, denoted as $\{\lambda_{i,j}^{z,w}\}_{i,j\in[N]}$ can be represented as

$$\lambda_{i,1}^{z,w} = \cdots = \lambda_{i,N-1}^{z,w} = 0, \lambda_{i,N}^{z,w} = -c_i, \quad i = 1, 2, \cdots, N-1,$$
$$\lambda_{N,j}^{z,w} = -d_j, \lambda_{N,N}^{z,w} = -\mathcal{Z}(1/c_N + 1/d_N), \quad j = 1, 2, \cdots, N-1.$$

The associated eigenvectors to $\lambda_{i,j}^{z,w}$, denoted as $\vec{u} = (\vec{u}_1, \cdots, \vec{u}_N)^\intercal$, satisfies

$$\begin{cases} \hat{R}_l^{z,w}\vec{u}_l + \hat{L}_l^{z,w}\vec{u}_N = \lambda_{i,j}^{z,w}\vec{u}_l, \quad l = 1, 2, \cdot, N-1 \\ \left(\hat{M}^{z,w} - \sum_l \hat{L}_l^{z,w}\right)\vec{u}_N = \lambda_{i,j}^{z,w}\vec{u}_N. \end{cases}$$

The eigenvectors can be studied in two cases:

(1) When $1 \leq i \leq N-1$, we can pick $\vec{u}_N = \mathbf{0}$. Then for $l = 1, 2, \cdots, N-1$, $\hat{R}_l^{z,w}\vec{u}_l = \lambda_{i,j}^{z,w}\vec{u}_l$. For $l \neq i$, we pick $\vec{u}_l = \mathbf{0}$. For $l = i$, $\vec{u}_i$ is the eigenvector to $\hat{R}_i^{z,w}$ associated with the eigenvalue $\lambda_{i,j}^{z,w}$: for $j = 1, 2, \cdots, N-1$, we pick $\vec{u}_i = \vec{u}_{i,j} = \vec{e}_j$. For $j = N$, we pick $\vec{u}_i = \vec{u}_{i,N}$ to be

$$\left(\frac{p(i,1)^{-w}p(i,1|z)^{1+w}}{\sum_l p(i,l)^{-w}p(i,l|z)^{1+w}}, \cdots, \frac{p(i,N-1)^{-w}p(i,N-1|z)^{1+w}}{\sum_l p(i,l)^{-w}p(i,l|z)^{1+w}}, -1\right)^\intercal \quad (27)$$

(2) When $i = N$, $\vec{u}_N \neq \mathbf{0}$. We need to solve $\left(\hat{M}^{z,w} - \sum_l \hat{L}_l^{z,w}\right)\vec{u}_N = \lambda_{i,j}^{z,w}\vec{u}_N$ first. For different $j$, we pick $\vec{u}_N = \vec{u}_{N,j}$ with $\vec{u}_{N,j} = \vec{e}_j$ for $j = 1, \cdots N-1$ and for $j = N$, $\vec{u}_{N,j} =$

$$\left(\frac{(\sum_l p(l,1))^{-w}(\sum_l p(l,1|z))^{1+w}}{-\lambda_{N,N}^{z,w} - d_1}, \cdots, \frac{(\sum_l p(l,N-1))^{-w}(\sum_l p(l,N-1|z))^{1+w}}{-\lambda_{N,N}^{z,w} - d_{N-1}}, -1\right)^\intercal \quad (28)$$

Next for each $j = 1, \cdots, N$, we solve $\left(\hat{R}_l^{z,w} - \lambda_{N,j}^{z,w}I_N\right)\vec{u}_{lj} = -\hat{L}_l^{z,w}\vec{u}_{N,j}$ for all $l = 1, 2, \cdots, N-1$. We get

$$\vec{u}_{l,j} = \begin{cases} -\dfrac{p(l,j)^{-w}p(l,j|z)^{1+w}}{\sum_{l'} p(l',j)^{-w}p(l',j|z)^{1+w}}\vec{e}_j, \quad j = 1, \cdots, N-1, \\ \left(\vec{u}_{l,N}(1), \cdots \vec{u}_{l,N}(N-1), \vec{u}_{l,N}(N)\right)^\intercal, j = N \end{cases} \quad (29)$$

with

$$\vec{u}_{l,N}(l') = -\frac{1}{\lambda_{N,N}^{z,w}}\left(\frac{1}{\lambda_{N,N}^{z,w} + c_l} + \frac{1}{\lambda_{N,N}^{z,w} + d_{l'}}\right)p(l,l')^{-w}p(l,l'|z)^{1+w},$$

$$\vec{u}_{l,N}(N) = -\frac{1}{c_l(\lambda_{N,N}^{z,w} + c_l)}\sum_{l'} p(l,l')^{-w}p(l,l'|z)^{1+w}.$$

Collect all the eigen information above, we diagonalize $\hat{Q}^{z,w}$ blockwisely: $\hat{Q}^{z,w} = \hat{X}^{z,w}\hat{D}^{z,w}(\hat{X}^{z,w})^{-1}$ s.t.

$$\hat{D}^{z,w} = \text{Diag}(\hat{D}_1^{z,w}, \cdots, \hat{D}_{N-1}^{z,w}, \hat{D}_N^{z,w}), \ \hat{D}_i^{z,w} = \text{Diag}(\lambda_{i,1}^{z,w}, \cdots \lambda_{i,N-1}^{z,w}, \lambda_{i,N}^{z,w}) \text{ for each } i$$

$$\hat{X}^{z,w} = \begin{pmatrix} \hat{X}_1^{z,w} & \mathbf{O} & \cdots & \mathbf{O} & -\hat{Y}_1^{z,w} \\ \mathbf{O} & \hat{X}_2^{z,w} & \cdots & \mathbf{O} & -\hat{Y}_2^{z,w} \\ \vdots & \vdots & \ddots & \vdots & \vdots \\ \mathbf{O} & \mathbf{O} & \cdots & \hat{X}_{N-1}^{z,w} & -\hat{Y}_{N-1}^{z,w} \\ \mathbf{O} & \mathbf{O} & \cdots & \mathbf{O} & \hat{X}_N^{z,w} \end{pmatrix},$$

$$(\hat{X}^{z,w})^{-1} = \begin{pmatrix} \hat{X}_1^{z,w} & \mathbf{O} & \cdots & \mathbf{O} & \hat{X}_1^{z,w}\hat{Y}_1^{z,w}\hat{X}_N^{z,w} \\ \mathbf{O} & \hat{X}_2^{z,w} & \cdots & \mathbf{O} & \hat{X}_2^{z,w}\hat{Y}_2^{z,w}\hat{X}_N^{z,w} \\ \vdots & \vdots & \ddots & \vdots & \vdots \\ \mathbf{O} & \mathbf{O} & \cdots & \hat{X}_{N-1}^{z,w} & \hat{X}_{N-1}^{z,w}\hat{Y}_{N-1}^{z,w}\hat{X}_N^{z,w} \\ \mathbf{O} & \mathbf{O} & \cdots & \mathbf{O} & \hat{X}_N^{z,w} \end{pmatrix},$$

where for each $i = 1, \cdots, N-1$,

$$\hat{D}_i^{z,w} = \mathrm{Diag}(0, \cdots, 0, -c_i),$$

$$\hat{X}_i^{z,w} = \begin{pmatrix} | & \cdots & | & | \\ \vec{e}_1 & \cdots & \vec{e}_{N-1} & \vec{u}_{i,N} \\ | & \cdots & | & | \end{pmatrix} = (\hat{X}_i^{z,w})^{-1} \text{ with } \vec{u}_{i,N} \text{ defined in equation 27}$$

$$\hat{Y}_i^{z,w} = -0 \begin{pmatrix} | & \cdots & | & | \\ \vec{u}_{i,1} & \cdots & \vec{u}_{i,N-1} & \vec{u}_{i,N} \\ | & \cdots & | & | \end{pmatrix} \text{ with } \{\vec{u}_{i,j}\}_{j=1}^N \text{ defined in equation 29},$$

and for $i = N$,

$$\hat{D}_N^{z,w} = \mathrm{Diag}(-d_1, \cdots, -d_{N-1}, -\mathcal{Z}^{z,w}/c_N - \mathcal{Z}^{z,w}/d_N),$$

$$\hat{X}_N^{z,w} = \begin{pmatrix} | & \cdots & | & | \\ \vec{e}_1 & \cdots & \vec{e}_{N-1} & \vec{u}_{N,N} \\ | & \cdots & | & | \end{pmatrix} = (\hat{X}_N^{z,w})^{-1} \text{ with } \vec{u}_{N,N} \text{ defined in equation 28}.$$

Step 3: solve the equation equation 10 explicitly The solution to equation 10 can be computed using the formula $q_t^{z,w} = \exp\left(\int_0^t \hat{Q}_{T-s}^{z,w} ds\right) q_0^{z,w}$, where the matrix $\exp\left(\int_0^t \hat{Q}_{T-s}^{z,w} ds\right)$ is computed using the eigenvalue decomposition in **Step 2**. More specifically,

$$\exp\left(\int_0^t \hat{Q}_{T-s}^{z,w} ds\right) = \exp\left(\int_0^t \frac{e^{-(T-s)}}{1-e^{-(T-s)}} ds \hat{Q}^{z,w}\right) = \hat{X}^{z,w} \exp\left(\ln\left(\frac{1-e^{-T}}{1-e^{-(T-t)}}\right)\hat{D}^{z,w}\right)(\hat{X}^{z,w})^{-1}$$

$$= \begin{pmatrix} \hat{X}_1^{z,w} & \mathbf{O} & \cdots & \mathbf{O} & -\hat{Y}_1^{z,w} \\ \mathbf{O} & \hat{X}_2^{z,w} & \cdots & \mathbf{O} & -\hat{Y}_2^{z,w} \\ \vdots & \vdots & \ddots & \vdots & \vdots \\ \mathbf{O} & \mathbf{O} & \cdots & \hat{X}_{N-1}^{z,w} & -\hat{Y}_{N-1}^{z,w} \\ \mathbf{O} & \mathbf{O} & \cdots & \mathbf{O} & \hat{X}_N^{z,w} \end{pmatrix} \mathrm{Diag}\left(\left(\frac{1-e^{-T}}{1-e^{-(T-t)}}\right)^{\hat{D}_1^{z,w}}, \cdots, \left(\frac{1-e^{-T}}{1-e^{-(T-t)}}\right)^{\hat{D}_N^{z,w}0}\right)$$

$$\begin{pmatrix} \hat{X}_1^{z,w} & \mathbf{O} & \cdots & \mathbf{O} & \hat{X}_1^{z,w}\hat{Y}_1^{z,w}\hat{X}_N^{z,w} \\ \mathbf{O} & \hat{X}_2^{z,w} & \cdots & \mathbf{O} & \hat{X}_2^{z,w}\hat{Y}_2^{z,w}\hat{X}_N^{z,w} \\ \vdots & \vdots & \ddots & \vdots & \vdots \\ \mathbf{O} & \mathbf{O} & \cdots & \hat{X}_{N-1}^{z,w} & \hat{X}_{N-1}^{z,w}\hat{Y}_{N-1}^{z,w}\hat{X}_N^{z,w} \\ \mathbf{O} & \mathbf{O} & \cdots & \mathbf{O} & \hat{X}_N^{z,w} \end{pmatrix}$$

$$= \begin{pmatrix} \hat{X}_1^{z,w}\left(\frac{1-e^{-T}}{1-e^{-(T-t)}}\right)^{\hat{D}_1^{z,w}}\hat{X}_1^{z,w} & \cdots & \mathbf{O} & \hat{M}_1^{z,w} \\ \vdots & \ddots & \vdots & \vdots \\ \mathbf{O} & \cdots & \hat{X}_{N-1}^{z,w}\left(\frac{1-e^{-T}}{1-e^{-(T-t)}}\right)^{\hat{D}_{N-1}^{z,w}}\hat{X}_{N-1}^{z,w} & \hat{M}_{N-1}^{z,w} \\ \mathbf{O} & \cdots & \mathbf{O} & \hat{X}_N^{z,w}\left(\frac{1-e^{-T}}{1-e^{-(T-t)}}\right)^{\hat{D}_N^{z,w}}\hat{X}_N^{z,w} \end{pmatrix}.$$

For each $i = 1, 2 \cdots, N-1$

$$\hat{X}_i^{z,w}\left(\frac{1-e^{-T}}{1-e^{-(T-t)}}\right)^{\hat{D}_1^{z,w}}\hat{X}_i^{z,w} = \begin{pmatrix} 1 & \cdots & 0 & \left(1 - \left(\frac{1-e^{-(T-t)}}{1-e^{-T}}\right)c_i\right)\frac{p^{z,w}(i,1)}{\sum_l p^{z,w}(i,l)} \\ \vdots & \ddots & \vdots & \vdots \\ 0 & \cdots & 1 & \left(1 - \left(\frac{1-e^{-(T-t)}}{1-e^{-T}}\right)c_i\right)\frac{p^{z,w}(i,N-1)}{\sum_l p^{z,w}(i,l)} \\ 0 & \cdots & 0 & \left(\frac{1-e^{-(T-t)}}{1-e^{-T}}\right)c_i \end{pmatrix},$$

$$\hat{M}_i^{z,w} := \hat{X}_i^{z,w}\left(\frac{1-e^{-T}}{1-e^{-(T-t)}}\right)^{\hat{D}_i^{z,w}}\hat{X}_i^{z,w}\hat{Y}_i^{z,w}\hat{X}_N^{z,w} - \hat{Y}_i^{z,w}\left(\frac{1-e^{-T}}{1-e^{-(T-t)}}\right)^{\hat{D}_N^{z,w}}\hat{X}_N^{z,w}$$

$$= \begin{pmatrix} \left(1 - \left(\frac{1-e^{-t}}{1-e^{-T}}\right)^{d_1}\right)\frac{p^{z,w}(i,1)}{\sum_l p^{z,w}(l,1)} & \cdots & 0 & \beta_{i,1}\mathcal{Z}p^{z,w}(i,1) \\ \vdots & \ddots & \vdots & \vdots \\ 0 & \cdots & \left(1 - \left(\frac{1-e^{-t}}{1-e^{-T}}\right)^{d_{N-1}}\right)\frac{p^{z,w}(i,N-1)}{\sum_l p^{z,w}(l,N-1)} & \beta_{i,N-1}\mathcal{Z}p^{z,w}(i,N-1) \\ 0 & \cdots & 0 & \beta_{i,N}\mathcal{Z}\sum_l p^{z,w}(i,l) \end{pmatrix},$$

where for each $i, j = 1, 2, \cdots, N-1$,

$$\beta_{i,j} := -\frac{1}{c_i(\lambda_{NN}^{z,w} + c_i)}\left(1 - \left(\frac{1-e^{-(T-t)}}{1-e^{-T}}\right)^{c_i}\right) - \frac{1}{d_j(\lambda_{NN}^{z,w} + d_j)}\left(1 - \left(\frac{1-e^{-(T-t)}}{1-e^{-T}}\right)^{d_j}\right)$$

$$- \frac{1}{\lambda_{NN}^{z,w}}\left(\frac{1}{\lambda_{NN}^{z,w} + c_i} + \frac{1}{\lambda_{NN}^{z,w} + d_j}\right)\left(1 - \left(\frac{1-e^{-(T-t)}}{1-e^{-T}}\right)^{-\lambda_{NN}^{z,w}}\right), \tag{30}$$

$$\beta_{i,N} := -\frac{1}{c_i(\lambda_{NN}^{z,w} + c_i)}\left(\left(\frac{1-e^{-(T-t)}}{1-e^{-T}}\right)^{c_i} - \left(\frac{1-e^{-(T-t)}}{1-e^{-T}}\right)^{-\lambda_{NN}^{z,w}}\right). \tag{31}$$

For $i = N$, we have

$$\hat{X}_N^{z,w}\left(\frac{1-e^{-T}}{1-e^{-(T-t)}}\right)^{\hat{D}_N^{z,w}}\hat{X}_N^{z,w}$$

$$= \begin{pmatrix} \left(\frac{1-e^{-(T-t)}}{1-e^{-T}}\right)^{d_1} & \cdots & 0 & \beta_{N,1}\mathcal{Z}\sum_l p^{z,w}(l,1) \\ \vdots & \ddots & \vdots & \vdots \\ 0 & \cdots & \left(\frac{1-e^{-(T-t)}}{1-e^{-T}}\right)^{d_{N-1}} & \beta_{N,N-1}\mathcal{Z}\sum_l p^{z,w}(l,N-1) \\ 0 & \cdots & 0 & \left(\frac{1-e^{-(T-t)}}{1-e^{-T}}\right)^{-\lambda_{NN}^{z,w}} \end{pmatrix}$$

where for each $j = 1, 2, \cdots, N-1$,

$$\beta_{N,j} := -\frac{1}{d_j(\lambda_{NN}^{z,w} + d_j)}\left(\left(\frac{1-e^{-(T-t)}}{1-e^{-T}}\right)^{d_j} - \left(\frac{1-e^{-(T-t)}}{1-e^{-T}}\right)^{-\lambda_{NN}^{z,w}}\right). \tag{32}$$

Now, we can apply the initial condition $q_0^{z,w} = \delta_{NN}$ to compute $q_t^{z,w}$.

$$q_t^{z,w} = \exp\left(\int_0^t \hat{Q}_{T-s}^{z,w}\mathrm{d}s\right)q_0^{z,w}$$

$$= \begin{pmatrix} \hat{X}_1^{z,w}\left(\frac{1-e^{-T}}{1-e^{-(T-t)}}\right)^{\hat{D}_1^{z,w}}\hat{X}_1^{z,w} & \cdots & \mathbf{O} & \hat{M}_1^{z,w} \\ \vdots & \ddots & \vdots & \vdots \\ \mathbf{O} & \cdots & \hat{X}_{N-1}^{z,w}\left(\frac{1-e^{-T}}{1-e^{-(T-t)}}\right)^{\hat{D}_{N-1}^{z,w}}\hat{X}_{N-1}^{z,w} & \hat{M}_{N-1}^{z,w} \\ \mathbf{O} & \cdots & \mathbf{O} & \hat{X}_N^{z,w}\left(\frac{1-e^{-T}}{1-e^{-(T-t)}}\right)^{\hat{D}_N^{z,w}}\hat{X}_N^{z,w} \end{pmatrix}\begin{pmatrix} \mathbf{0} \\ \vdots \\ \mathbf{0} \\ \vec{e}_N \end{pmatrix}$$

$$= \begin{pmatrix} \hat{M}_1^{z,w}(:,N) \\ \vdots \\ \hat{M}_{N-1}^{z,w}(:,N) \\ (\hat{X}_N^{z,w}\left(\frac{1-e^{-T}}{1-e^{-(T-t)}}\right)^{\hat{D}_N^{z,w}}\hat{X}_N^{z,w})(:,N) \end{pmatrix}.$$

Therefore, for all $i, j = 1, 2\cdots, N-1$,

$$q_t^{z,w}(i,j) = \hat{M}_i^{z,w}(j,N) = \beta_{i,j}\mathcal{Z}p^{z,w}(i,j).$$

For $j = N, i = 1, 2, \cdots, N-1$,

$$q_t^{z,w}(i,N) = \hat{M}_i^{z,w}(N,N) = \beta_{i,N}\mathcal{Z}\sum_l p^{z,w}(i,l).$$

For $i = N, j = 1, 2, \cdots, N-1$,

$$q_t^{z,w}(N,j) = \beta_{N,j}\mathcal{Z}\sum_l p^{z,w}(l,j).$$

Last, for $i = j = N$,

$$q_t^{z,w}(N,N) = \left(\frac{1-e^{-(T-t)}}{1-e^{-T}}\right)^{-\lambda_{NN}^{z,w}}.$$

Last, Theorem 3.2 follows from the following definition of $\alpha_t := \beta_{x_1,x_2}$ for all $x \in \{1, 2, \cdots, N\}^2$.

$\square$

### E.1 SAMPLED DISTRIBUTIONS FOR $D = 2$

*Proof of Proposition 3.3.* According to equation 15 and Assumption 1.1, we have

$$q_T^{z_1,w}(x) \propto (1/c_{x_1} + 1/d_{x_2})p(x)^{-w}p(x|z_1)^{1+w}$$

$$\propto \left( \underbrace{\left( \frac{a_1 p(x_1|z_1)}{\sum_k a_k p(x_1|z_k)} \right)^w}_{\text{I}} + \underbrace{\left( \frac{a_1 p(x_2|z_1)}{\sum_k a_k p(x_2|z_k)} \right)^w}_{\text{II}} \right) \underbrace{\left( \frac{a_1 p(x|z_1)}{\sum_k a_k p(x|z_k)} \right)^w}_{\text{III}} p(x|z_1).$$

Each of the terms I, II and III is within the range $[0, 1]$ and exponentially dependent to $w$. Therefore, the values of I, II and III affect the sampled distribution significantly when $w$ is large. By evaluating I, II and III in different regions depending on relations between the marginal supports, we express $q_T^{z_1,w}$ as presented in Proposition 3.3. The last statement in Proposition 3.3 follows from the **discussion on** $q^{z_1,\infty}(\cdot|z_1)$ in this section. $\square$

**Effect of guidance on sampled distributions.** According to Proposition 3.3-(2), $q_T^{z_1,w}$ is defined with different weight-adjustment in 5 different type of regions defined as follows

$$\begin{aligned}
\mathcal{R}_1 &\coloneqq \{x|x \in \mathcal{X}_1, x_1 \in \mathcal{X}_{1,1} \setminus S_{1,1}, x_2 \in \mathcal{X}_{1,2} \setminus S_{1,2}\}, \\
\mathcal{R}_{2,i} &\coloneqq \{x|x \in \mathcal{X}_1, x_i \in S_{1,i}, x_{\setminus i} \in \mathcal{X}_{1,\setminus i} \setminus S_{1,\setminus i}\}, \qquad i = 1, 2, \\
\mathcal{R}_3 &\coloneqq \{x|x \in \mathcal{X}_1 \setminus S_1, x_1 \in S_{1,1}, x_2 \in S_{1,2}\}, \\
\mathcal{R}_4 &\coloneqq S_1.
\end{aligned}$$

The above sets reflect different level of "privacy" of class $z_1$. $\mathcal{R}_4$ is the shared region with other classes. $\mathcal{R}_1, \mathcal{R}_{2,i}, \mathcal{R}_3$ are not shared with other classes. But $\mathcal{R}_3$ has both marginals shared with other classes and $\mathcal{R}_{2,i}$ has one of the marginals shared with other classes. $\mathcal{R}_1$ is the most private set in class $z_1$, with no intersection with other classes even for marginals. The associated weights (before normalization) on different regions are give by:

$$A^{z_1,w} = 2,$$

$$A_{2,i}^{z_1,w} = 1 + \left( \frac{a_1 p(x_i|z_1)}{\sum_{k \in I_{1,i}} a_k p(x_i|z_k)} \right)^w, \qquad i = 1, 2,$$

$$A_3^{z_1,w} = \sum_{i=1}^2 \left( \frac{a_1 p(x_i|z_1)}{\sum_{k \in I_{1,i}} a_k p(x_i|z_k)} \right)^w,$$

$$A_4^{z_1,w} = \left( \sum_{i=1}^2 \left( \frac{a_1 p(x_i|z_1)}{\sum_{k \in I_{1,i}} a_k p(x_i|z_k)} \right)^w \right) \left( \frac{a_1 p(x|z_1)}{\sum_{k \in I_1} a_k p(x|z_k)} \right)^w,$$

we can notice that for all $w \geq 0$, $A_1^{z_1,w} \geq A_{2,i}^{z_1,2} \geq A_3^{z_1,w} \geq A_4^{z_1,w}$. This reflects that *the sampled distribution from the discrete diffusion with CFG can leverage the geometric information of the full data distribution: the sampled distribution puts larger weights on more private regions of class $z_1$.* We conjecture that the above fact is also true in high dimension:

**Conjecture E.1.** *For any $D \geq 2$, discrete diffusion with CFG leverages the geometric information from the full data distribution. More specifically, under Assumption 1.1, the sampled distribution $q_T^{z_1,w}$ adapts the class distribution $p(\cdot|z_1)$ by putting larger weights on more private regions of class $z_1$, where those regions with different privacy are defined based on the support sets and their marginals.*

**Discussion on** $q^{z_1,\infty}(\cdot|z_1)$**.** Now we look at the structure of the sampled distribution as $w \to \infty$ in further detail. According to the expression of weights $A^{z_1,w}$, since $1 \in I_{1,i}$ for $i = 1, 2$ and $1 \in I_1$, the rational factors inside the parentheses is in $(0, 1]$. In particular, if $S_1 \neq \emptyset$, i.e., class $z_1$ has intersected domain with other classes, $|I_1| \geq 2$. Hence $\frac{a_1 p(x|z_1)}{\sum_{k \in I_1} a_k p(x|z_k)} \in (0, 1)$. Therefore, as $w \to \infty$, we have

$$A_1^{z_1,\infty} = 2, \quad A_{2,i}^{z_1,\infty} \in \{1, 2\}, \quad A_3^{z_1,\infty} \in \{0, 1, 2\}, \quad A_4^{z_1,\infty} = 0.$$

Then, we have $q^{z_1,\infty}(\cdot|z_1)|_{A_4^{z_1,\infty}} = 0$, i.e., $\mathrm{Supp}(q^{z_1,\infty}(\cdot|z_1)) \subset \mathcal{X}_1 \setminus S_1$.

It is worth noting that it is possible that some sets among $\mathcal{R}_1, \mathcal{R}_{2,i}, \mathcal{R}_3$ could be empty. Therefore, for a general data distribution $p$ satisfying Assumption 1.1, in order to derive $q^{z_1,\infty}(\cdot|z_1)$ completely, we need to first identity whether $\mathcal{R}_1, \mathcal{R}_{2,i}, \mathcal{R}_3$ are non-empty or not, and then compute the associated limiting weights on the non-empty regions. In the following, we will use a simple example to illustrate this procedure.

*An example with $D = 2, N = 5$.* We consider the data distribution $p$ is a mixture of two classes with equal weights: $p(x) = \frac{1}{2}p(x|z_1) + \frac{1}{2}p(x|z_2)$ for all $x \in \{1, 2, 3, 4, 5\}^2$ with 5 being the masked state. The heat maps for $p(\cdot|z_1), p(\cdot|z_2)$ and $p$ are given in Figure 4. We can distinguish the regions

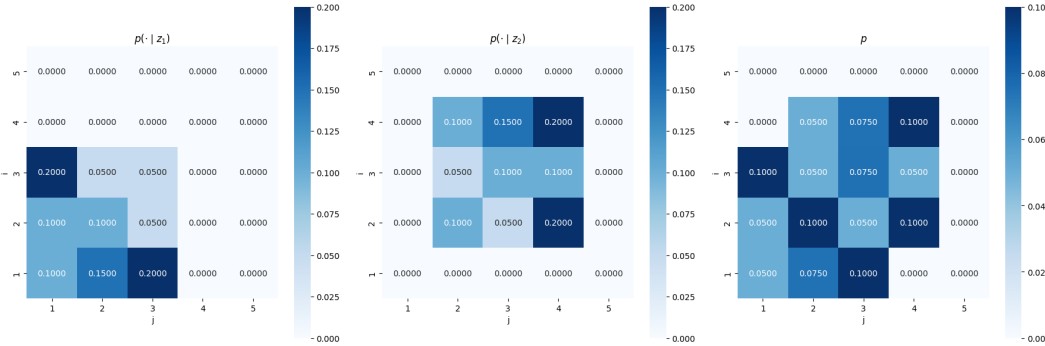

Figure 4: heat maps for $p(\cdot|z_1), p(\cdot|z_2)$ and $p$.

with different level of privacy based on our formulas. As shown in Figure 5, we notice that $\mathcal{R}_3 = \emptyset$ and $\mathcal{R}_1, \mathcal{R}_{2,1}, \mathcal{R}_{2,2}, \mathcal{R}_4$ are identified with different colors. Based on the information of $p$, we can

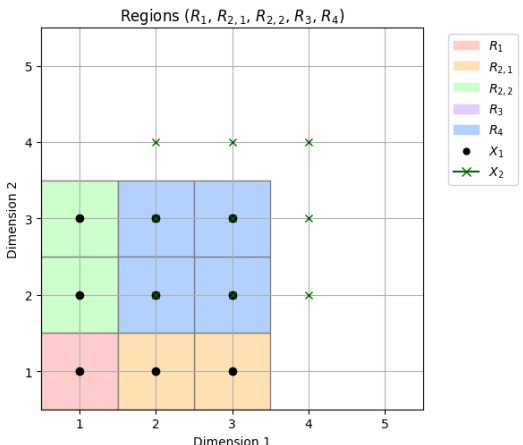

Figure 5: identification of different regions.

compute the limiting weights as $w \to \infty$. We have

$$A_1^{z_1,\infty} = 2, \quad A_{2,1}^{z_1,\infty} = 1_{x_1=1}, \quad A_{2,2}^{z_1,\infty} = 1_{x_2=1}, \quad A_4^{z_1,\infty} = 0.$$

Therefore, the sampled distribution $q_T^{z_1,\infty}(\cdot|z_1)$ adapts $p(\cdot|z_1)$ by putting these weights on the 4 regions respectively, i.e.,

$$q_T^{z_1,\infty}(x|z_1) \propto \begin{cases} 2p(x|z_1), & x \in \mathcal{R}_1 = \{(1,1)\}, \\ p(x|z_1), & x \in \mathcal{R}_{2,1} = \{(2,1),(3,1))\}, \\ p(x|z_1), & x \in \mathcal{R}_{2,2} = \{(1,2),(1,3))\}, \\ 0, & \text{otherwise}, \end{cases}$$

which implies that $q_T^{z_1,\infty}(1,1|z_1) = q_T^{z_1,\infty}(1,3|z_1) = q_T^{z_1,\infty}(3,1|z_1) = 4/17$, $q_T^{z_1,\infty}(1,2|z_1) = q_T^{z_1,\infty}(2,1|z_1) = 3/17$ and $q_T^{z_1,\infty}(x_1,x_2|z_1) = 0$ otherwise. In Figure 6, we present the heatmaps for the class distribution of $z_1$, the tilted distributions and the sampled distributions with $w = 1, 5, 15$. We can observe the following facts that match our theory.

(1) the sampled distribution deviates from the tilted distribution for all $w > 0$.

(2) the effects of guidance differ in different regions: as $w$ increases, the probability mass decreases in $S_1 = \{(2,2),(2,3),(3,2),(3,3)\}$; the probability mass increases in regions $\mathcal{R}_{2,1} = \{(2,1),(3,1)\}$ and $\mathcal{R}_{2,2} = \{(1,2),(1,3)\}$ at the same rate; the probability mass increases in the region $\mathcal{R}_1 = \{1,1\}$ at the largest rate.

(3) for large guidance ($w = 15$), the sampled distribution $q_T^{z_1,w}$ can be approximately understood as $q_T^{z_1,\infty}(\cdot|z_1)$. The last plot in Figure 6 matches our computation for $q_T^{z_1,\infty}(\cdot|z_1)$.

(4) for small guidance ($w = 1$), the effect of guidance is also small. The sampled distribution $q_T^{z_1,w}$ deviates a little bit from the target distribution $p(\cdot|z_1)$ in the way we described in (2).

In practice, people observe that the optimal guidance is usually positive but small (of order $\Theta(1)$). Our theory and numerical observations bring insights in understanding the optimal guidance. Roughly speaking, if we can show that the effects of guidance presented above actually compensate the effect of score approximation, by quantifying the inductive bias in learning the scores, we can rigorously analyze the optimal guidance in the CFG setting. This will be left as an interesting future work to explore.

### E.2 CONVERGENCE RATES FOR $D = 2$

*Proof of Proposition 3.4.* For simplicity, we denote $\lambda := \lambda_{N,N}^{z,w}$. According to Theorem 3.2 and Remark 3.3, the total variation distance can be computed as

$$
\begin{aligned}
&\text{TV}(q_t^{z,w}, q_T^{z,w})\\
&= \frac{1}{2} \sum_{x \in S} |q_t^{z,w}(x) - q_T^{z,w}(x)|\\
&= \frac{1}{2} \sum_{x \neq (N,N)} |\alpha_t(x) - \alpha_T(x)| \mathcal{Z} p^{z,w}(x) + \frac{1}{2} |\alpha_t(N,N) - \alpha_T(N,N)|\\
&= \frac{1}{2} \mathcal{Z} \sum_{x_1,x_2 \neq N} p^{z,w}(x) \left( |\frac{1}{c_{x_1} + \lambda}(\frac{1}{c_{x_1}} r(t)^{c_{x_1}} + \frac{1}{\lambda} r(t)^{-\lambda})| + |\frac{1}{d_{x_2} + \lambda}(\frac{1}{d_{x_2}} r(t)^{d_{x_2}} + \frac{1}{\lambda} r(t)^{-\lambda})| \right)\\
&+ \frac{1}{2} \mathcal{Z} \sum_{x_1 \neq N} p^{z,w}(x_1) |\frac{1}{c_{x_1}(\lambda + c_{x_1})}(r(t)^{c_{x_1}} - r(t)^{-\lambda})|\\
&+ \frac{1}{2} \mathcal{Z} \sum_{x_2 \neq N} p^{z,w}(x_2) |\frac{1}{d_{x_2}(\lambda + d_{x_2})}(r(t)^{d_{x_2}} - r(t)^{-\lambda})| + \frac{1}{2} r(t)^{-\lambda}\\
&:= \text{I} + \text{II} + \text{III} + \text{IV},
\end{aligned}
$$

where $r(t) := \frac{1-e^{-(T-t)}}{1-e^{-T}} \in (0,1)$. Next, we bound each term respectively.

For I, we bound the two terms inside using the following properties of function $h_1 : y \in [1,\infty) \mapsto y^{-1} r(t)^y$: $h_1'(y) < 0$ and $h_1''(y) > 0$ for all $y$. Notice that $c_l, d_l \geq 1$ and $-\lambda = \frac{\mathcal{Z}}{c_N} + \frac{\mathcal{Z}}{d_N} = \sum_{l_1} p(l_1)^{-w} p(l_1|z)^{1+w} + \sum_{l_2} p(l_2)^{-w} p(l_2|z)^{1+w} = \exp(w\mathcal{D}_{1+w}(p_1(\cdot|z)\|p_1(\cdot))) + \exp(w\mathcal{D}_{1+w}(p_2(\cdot|z)\|p_2(\cdot))) \geq 2$, where we use $\mu_i$ to represent the $i^{th}$ marginal of $\mu$. Therefore, we have

$$
|\frac{1}{c_{x_1} + \lambda}(\frac{1}{c_{x_1}} r(t)^{c_{x_1}} + \frac{1}{\lambda} r(t)^{-\lambda})| = |\frac{h(c_{x_1}) - h(-\lambda)}{c_{x_1} - (-\lambda)}| = -h'(c_{x_1}^*) = \frac{1}{c_{x_1}^*} r(t)^{c_{x_1}^*}(\frac{1}{c_{x_1}^*} - \ln r(t))
$$

$$
|\frac{1}{d_{x_2} + \lambda}(\frac{1}{d_{x_2}} r(t)^{d_{x_2}} + \frac{1}{\lambda} r(t)^{-\lambda})| = |\frac{h(d_{x_2}) - h(-\lambda)}{d_{x_2} - (-\lambda)}| = -h'(d_{x_2}^*) = \frac{1}{d_{x_2}^*} r(t)^{d_{x_2}^*}(\frac{1}{d_{x_2}^*} - \ln r(t)),
$$

where $c_{x_1}^*$ is between $c_{x_1}$ and $-\lambda$, $d_{x_2}^*$ is between $d_{x_2}$ and $-\lambda$.

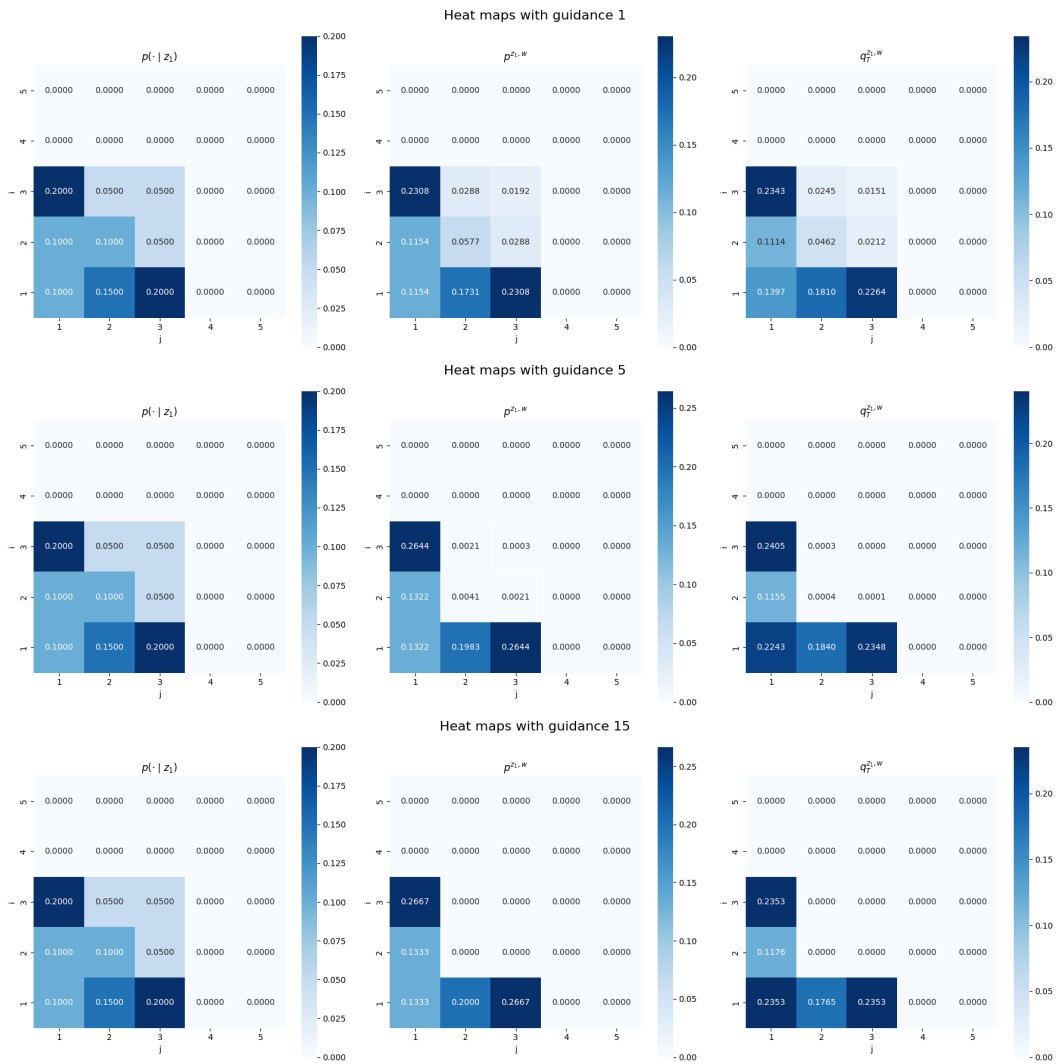

Figure 6: distributions under different guidance strengths: $w = 1, 5, 15$. The first column presents the class distribution of $z_1$. The second column presents the tilted distributions. The third column presents the sampled distributions which are obtained using exact evaluations of scores and integrals.

For II and III, we bound the two terms using the property of the function $h_2 : y \in [1, \infty) \mapsto r(t)^y$: $h_2'(y) < 0$ and $h_2''(y) > 0$ for all $y$. Again, due to the fact that $c_l, d_l \geq 1$ for all $l$ and $-\lambda \geq 2$, we have

$$\left| \frac{1}{c_{x_1}(\lambda + c_{x_1})} \left( r(t)^{c_{x_1}} - r(t)^{-\lambda} \right) \right| = \frac{1}{c_{x_1}} \left| \frac{h_2(c_{x_1}) - h_2(-\lambda)}{c_{x_1} - (-\lambda)} \right| = -\frac{1}{c_{x_1}} h_2'(c_{x_1}') = -\frac{1}{c_{x_1}} r(t)^{c_{x_1}'} \ln r(t)$$

$$\left| \frac{1}{d_{x_2}(\lambda + d_{x_2})} \left( r(t)^{d_{x_2}} - r(t)^{-\lambda} \right) \right| = \frac{1}{d_{x_2}} \left| \frac{h_2(d_{x_2}) - h_2(-\lambda)}{d_{x_2} - (-\lambda)} \right| = -\frac{1}{d_{x_2}} h_2'(d_{x_2}') = -\frac{1}{d_{x_2}} r(t)^{d_{x_2}'} \ln r(t),$$

where $c_{x_1}'$ is between $c_{x_1}$ and $-\lambda$, $d_{x_2}'$ is between $d_{x_2}$ and $-\lambda$.

Last, according to the expression of $c_l, d_l$ in equation 25, we have

$$c_l = \sum_{l'} \left( \frac{p(x_2 = l'|x_1 = l, z)}{p(x_2 = l'|x_1 = l)} \right)^{w+1} p(x_2 = l'|x_1 = l) = \exp\left( w \mathcal{D}_{1+w}(p(\cdot|x_1 = l, z) | p(\cdot|x_1 = l)) \right),$$

$$d_l = \sum_{l'} \left( \frac{p(x_1 = l'|x_2 = l, z)}{p(x_1 = l'|x_2 = l)} \right)^{w+1} p(x_1 = l'|x_2 = l) = \exp\left( w \mathcal{D}_{1+w}(p(\cdot|x_2 = l, z) | p(\cdot|x_2 = l)) \right).$$

For $w \gg 1$, since $c^*_{x_1}, c'_{x_1}$ are between between $c_{x_1}$ and $-\lambda$ and $\ln c_{x_1} = \Theta(w), \ln(-\lambda) = \Theta(w)$, we have $c^*_{x_1} = \Theta(w), c'_{x_1} = \Theta(w)$ for all $x_1$. For the same reason, $d^*_{x_2} = \Theta(w), d'_{x_2} = \Theta(w)$ for all $x_2$. Therefore, if we focus on the order of $w$ for $w \gg 1$ and preserve the leading order terms in $\text{TV}(q_t^{z,w}, q_T^{z,w})$, we have

$$\text{TV}(q_t^{z,w}, q_T^{z,w}) = \mathcal{Z} \exp(-\Theta(w)) r(t)^{\exp(\Theta(w))} + r(t)^{\exp(\Theta(w))}$$
$$= \exp(\Theta(w)) \exp(-\Theta(w)) r(t)^{\exp(\Theta(w))} + r(t)^{\exp(\Theta(w))},$$

where the second identity follows from Remark 3.2. $\qquad \square$

## F NUMERICAL EXPERIMENTS

### F.1 DETAILS ON 1D EXPERIMENT

We consider each cluster to be defined by the following density vector:

$$(0.1, 0.2, 0.4, 0.2, 0.1)$$

The full distribution contains two classes with equal weights, each class containing two of the clusters above.

**Disjoint Example:** we plot the class distributions and full probability distribution for the disjoint example in Figure 7.

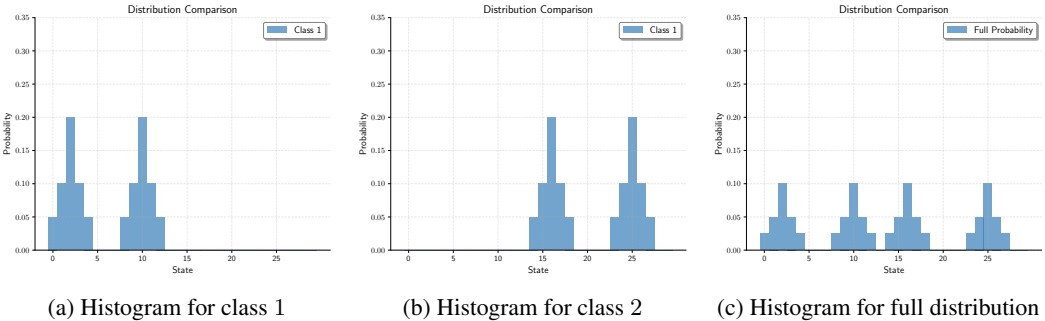

(a) Histogram for class 1     (b) Histogram for class 2     (c) Histogram for full distribution

Figure 7: Histograms corresponding to the disjoint example.

**Overlapping Example:** we pull the classes together to create a overlapping region. We plot the class conditional and full probability distributions for the overlapping example in Figure 8.

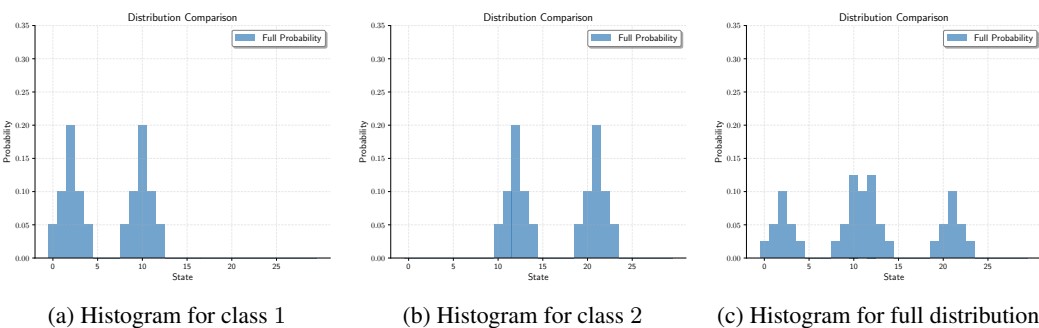

(a) Histogram for class 1     (b) Histogram for class 2     (c) Histogram for full distribution

Figure 8: Histograms corresponding to the overlapping example.

### F.2 Details on 2D Experiment

**Disjoint Example:** we plot the class distributions and full probability distribution for the disjoint example in Figure 9.

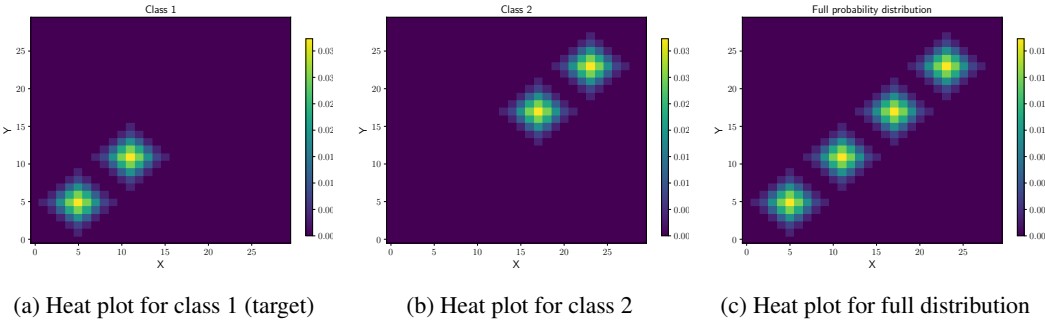

(a) Heat plot for class 1 (target)     (b) Heat plot for class 2     (c) Heat plot for full distribution

Figure 9: Heat plot corresponding to the disjoint example.

**Overlapping Example:** we pull the classes together to create intersection between the central diamond regions. We plot the class distributions and full probability distribution for the overlapping example in Figure 10.

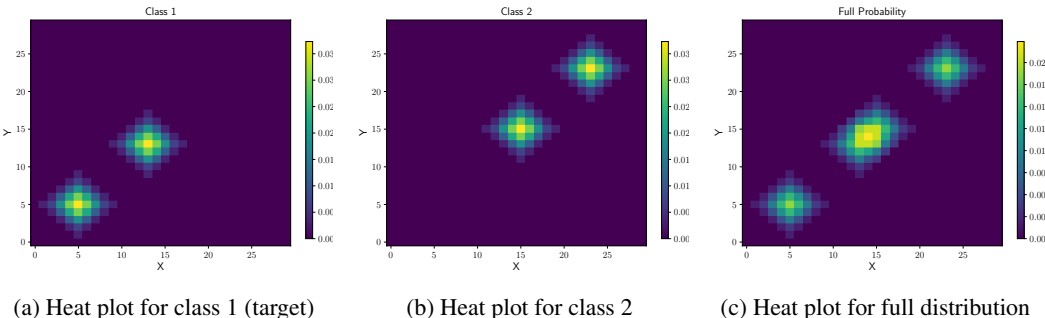

(a) Heat plot for class 1 (target)     (b) Heat plot for class 2     (c) Heat plot for full distribution

Figure 10: Heat plot corresponding to the overlapping example.

**Total variation as a function of** $w$**:** in Figure 17, we plot the total variation between intermediate generated distribution $q_t^{z,w}$ along the reverse dynamics at $t = .5$ and the final generated distribution $q_T^{z,w}$ as a function of $w$. We did not observe an exponential-decay when $w$ is large, which does not match results in Proposition 3.4. As shown in Theorem 3.2, when $w$ gets larger, the reserve dynamics admits sharper transitions, making the numerical method less efficient. Therefore, we conjecture that this mismatch comes from the propagation of the numerical error along the reverse process, and halfway through the simulation, the numerical error dominates the TV-dynamics.

### F.3 Experiments in 5D

**Marginals of distributions:** our full distribution is mixtures of uniform hypercubes, $\{0, 1, 2\}^5$ and $\{2, 3, 4\}^5$, which overlap only at the single state $(2, 2, 2, 2, 2)^\top$, and the target class is the one supported on $\{2, 3, 4\}^5$. The 2-dimensional marginals of the full distribution, class 0 and class 1 (target class) are plotted in Figures 11, 12 and 13. As introduced in Section 4, the geometry naturally partitions the space by the number of coordinates equal to 2, denoted by $k = \#\{d : x_d = 2\}$. The case $k = 0$ corresponds to states unique to the target class, $k = 5$ corresponds to the fully overlapping state, and intermediate values $k \in \{1, 2, 3, 4\}$ represent partially overlapping regions.

**Experiment setting:** we generate 10K samples, compute samples statistics (in Tables 1 and 2) and plot target class density ratios from the marginal of the generated distribution $q_T^{z,w}$ to the marginal

of the target class distribution $p(\cdot|z)$ with guidance $w = 1, 2, 4$ in Figures 14, 15 and 16 respectively. Our results show the generated 2-dimensional marginals also exhibit a similar structure to the generated distribution in 2D, that unique region has the largest per-state mass and the fully overlapping region has the smallest per-state mass. As $w$ increases, mass is shifted away from regions that are more ambiguous (large $k$) to regions that are more unique to the target class (small $k$). These numerical results support our discussion in Section 3.3.

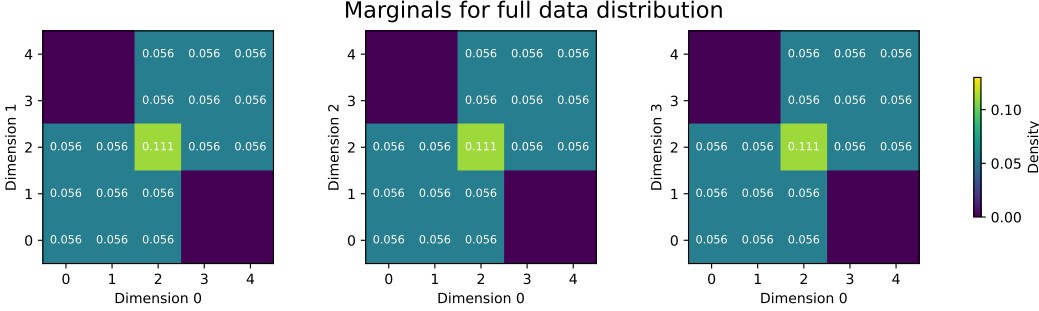

Figure 11: Marginals for the full data distribution

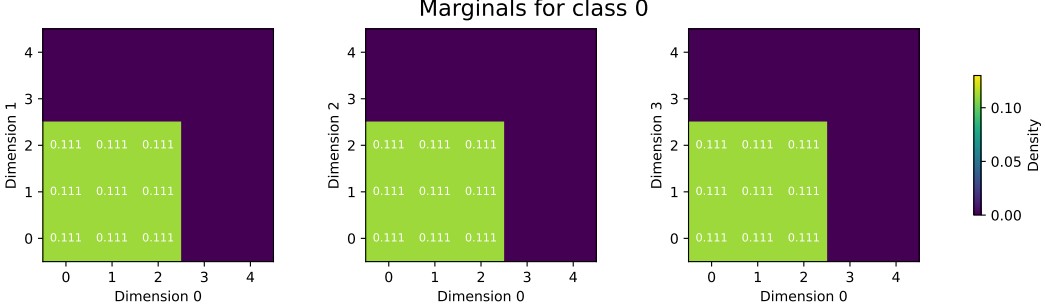

Figure 12: Marginals for class-0

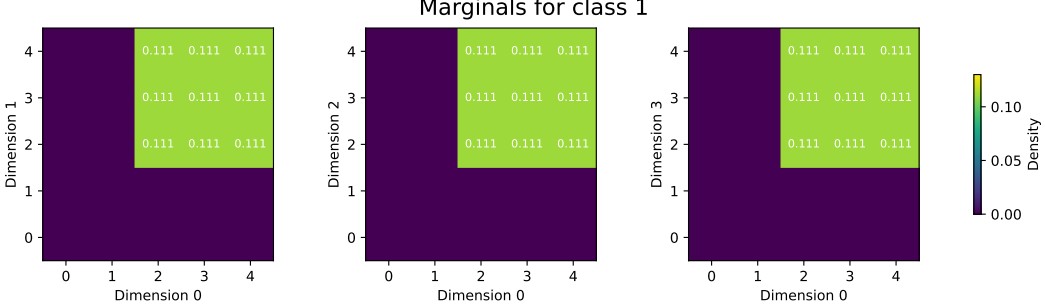

Figure 13: Marginals for class-1

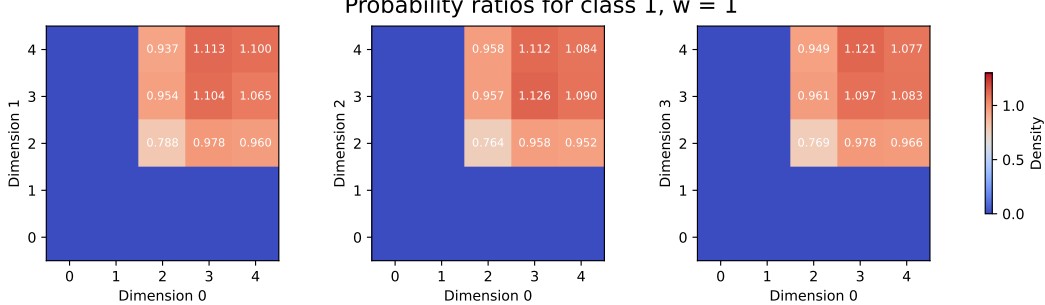

Figure 14: Probability ratios from the marginal of the generated distribution $q_T^{z,w}$ to the marginal of the target class distribution $p(\cdot|z)$ with $w = 1$.

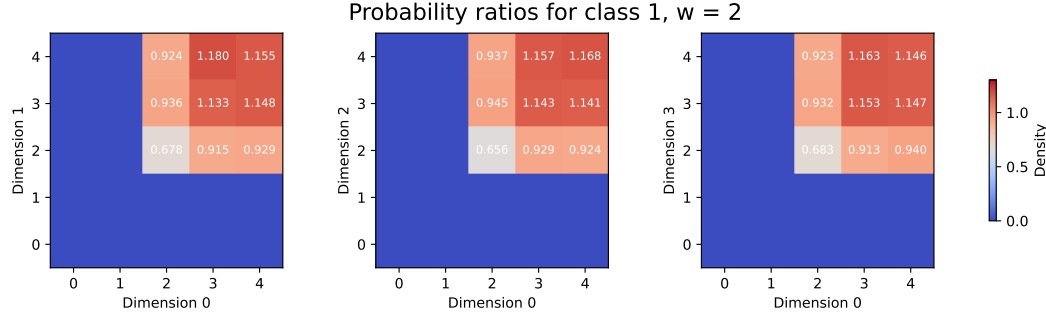

Figure 15: Probability ratios from the marginal of the generated distribution $q_T^{z,w}$ to the marginal of the target class distribution $p(\cdot|z)$ with $w = 2$.

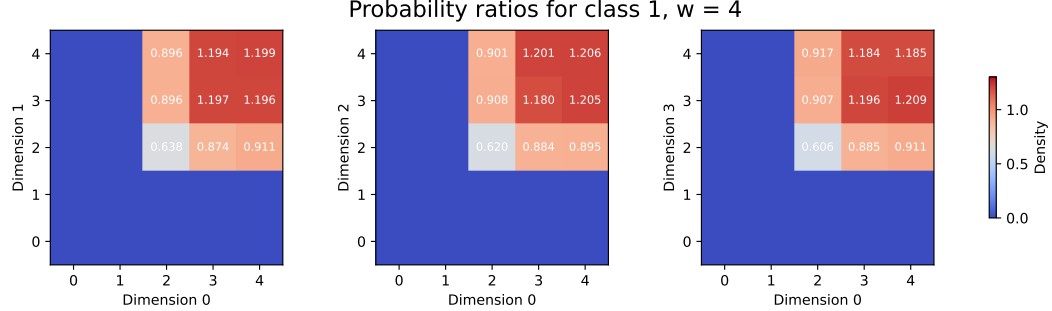

Figure 16: Probability ratios from the marginal of the generated distribution $q_T^{z,w}$ to the marginal of the target class distribution $p(\cdot|z)$ with $w = 4$.

## F.4 EXPERIMENTS ON MNIST

We demonstrate that our findings apply in high dimensional problems and practical settings. We trained a U-ViT network (Bao et al., 2023) for 100K iterations using the Adam optimizer with $1e-4$ learning rate. The hyperparameters for the network can be found in Figure 18.

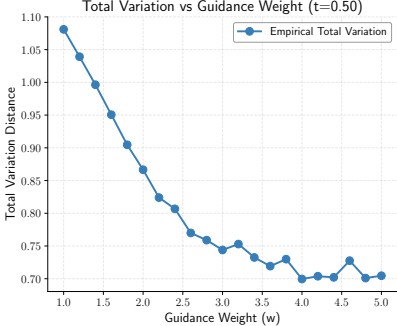

Figure 17: $\text{TV}(q_t^{z,w}, q_T^{z,w})$ as a function of $w$ with $t = .5$.

| Parameter | Value |
|---|---|
| **img_size** | 28 |
| **in_chans** | 1 |
| **patch_size** | 2 |
| **embed_dim** | 512 |
| **depth** | 12 |
| **num_heads** | 8 |
| **mlp_ratio** | 4 |
| **qkv_bias** | False |
| **mlp_time_embed** | False |
| **labels_dim** | 11 |

Figure 18: Model Configuration

We illustrate how guidance suppresses regions of intersection between classes through two case studies. In the first, we generate samples of digit 8 under three conditions: without guidance, with guidance, and with guidance using digit 3 as the conditioning class. As shown in Figure 19, guidance progressively removes ambiguous samples of 8 that resemble digit 3, with the effect being especially pronounced when digit 3 is used as the guiding distribution. A similar phenomenon is observed when generating digit 7 conditioned on digit 1 (Figure 20). Together, these results provide empirical evidence that the suppression of overlapping regions—predicted by our theoretical analysis in low dimensions—extends to high-dimensional, practical settings such as image generation.

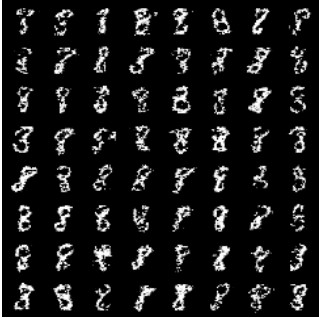 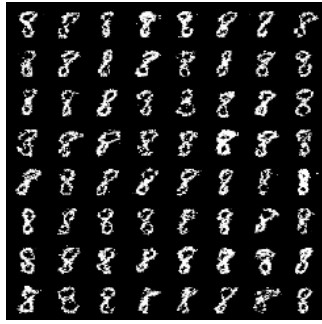 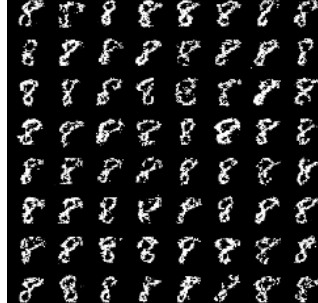

(a) Generating samples of 8 with no guidance

(b) Generating samples of 8 using guidance with $w = 1$

(c) Generating samples of 8 but using the class of number 3 as the guiding distribution using $w = 1$

Figure 19: Applying guidance reduces the are of intersection between classes. Notice how number 8's that look similar to a 3 disappear.

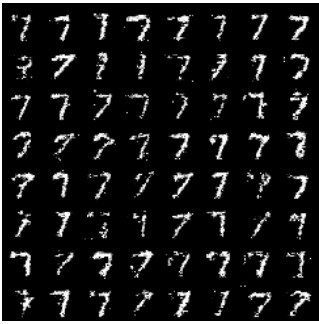 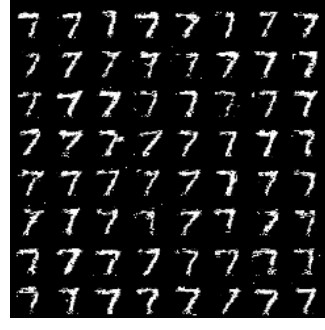 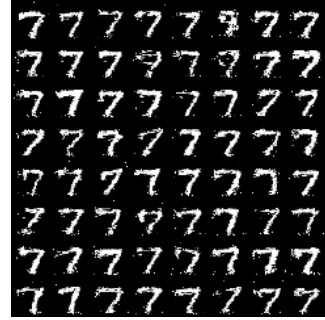

(a) Generating samples of 7 with no guidance

(b) Generating samples of 7 using guidance with $w = 2$

(c) Generating samples of 7 but using the class of number 1 as the guiding distribution using $w = 2$

Figure 20: Applying guidance reduces the area of intersection between classes. Notice how number 7's that resemble a 1 disappear.

## STATEMENT ON THE USE OF LARGE LANGUAGE MODELS

This work made use of large language models to assist with proofreading and improving the clarity of the writing. All research ideas, theoretical development, and experiments were carried out solely by the authors. When used for coding, it was solely used for plotting purposes.

