# OpenReview forum: "What Exactly Does Guidance Do in Masked Discrete Diffusion Models"
_ICLR.cc/2026/Conference — ICLR 2026 Poster_

### Official Review · Reviewer_XFWx · 2025-10-20

**Soundness:** 4
**Presentation:** 4
**Contribution:** 3
**Rating:** 8
**Confidence:** 4

**Summary:**

This paper provides a theoretical analysis of classifier-free guidance (CFG) in masked discrete diffusion models. In 1D, the guided reverse process exactly recovers the tilted distribution. In 2D, deviations emerge: the final distribution is a reweighted version of the tilted distribution, with mass suppressed even in regions overlapping only in projection.

The authors derive closed-form expressions in both settings and show that convergence to the final distribution exhibits double-exponential decay in the guidance strength w. Experiments in higher dimensions (5D, MNIST) confirm that these geometric effects—amplifying private regions and suppressing ambiguous ones—persist beyond low-dimensional cases.

**Strengths:**

I like this work. The paper is well-written and clear. It presents a novel theoretical insight into how classifier-free guidance (CFG) affects discrete diffusion sampling, with concrete examples and rigorous proofs. The authors also provide high-dimensional experiments to support the generality of their findings.

**Weaknesses:**

It would be interesting to discuss how this effect manifests in practical applications, and how the misalignment introduced by CFG could be addressed in real-world settings.

**Questions:**

See weakness.

---

> ### Author Response · Authors · 2025-11-21
>
> We thank the reviewer for the positive assessment and constructive feedback. We address the mentioned weaknesses below.
>
> >It would be interesting to discuss how this effect manifests in practical applications, and how the misalignment introduced by CFG could be addressed in real-world settings.
>
> This is a thoughtful suggestion. Our analysis shows that classifier free guidance concentrates probability on class specific regions and suppresses mass in regions of class overlap, which can lead to misalignment between the guided distribution and the true conditional distribution in practical applications. The extent of this effect depends on the geometry of the data and the intrinsic variability of the target class.
>
> Mitigating such misalignment would require compensating for the redistribution induced by guidance. One possible approach is to incorporate regularization during training that preserves probability mass in overlapping regions, counteracting the suppression applied during sampling. Another possibility is to adjust the effective guidance level at inference time using calibration procedures informed by geometric structure in the data. Although we do not explore these directions here, our theoretical results offer a framework for developing methods that control or correct misalignment in real world systems.

---

### Official Review · Reviewer_wYwP · 2025-10-31

**Soundness:** 2
**Presentation:** 2
**Contribution:** 3
**Rating:** 6
**Confidence:** 3

**Summary:**

This paper develops the first rigorous quantitative theory explaining the role of classifier-free guidance (CFG) in masked discrete diffusion models.
The authors analyze low-dimensional cases (1D and 2D) where the reverse dynamics can be solved exactly. Their results show that:
- Guidance amplifies class-specific regions while suppressing overlapping regions between classes, with overlap vanishing as guidance strength $w$ increases.
- In 1D, the generated distribution exactly matches the tilted distribution $p_{z,w} \propto p(x)p(z|x)^{1+w}$.
- In 2D, deviations emerge but can be expressed in closed form via coupling coefficients $c_x,d_x$.
- The convergence rate of the reverse dynamics exhibits a double-exponential dependence on w in both 1D and 2D.
Empirical illustrations confirm the theoretical predictions.

**Strengths:**

- Novel theoretical framework: First rigorous analysis of discrete CFG dynamics.
- Analytic tractability: Closed-form results for both 1D and 2D masked diffusion.
- Clear phenomena: Demonstrates class-specific amplification and overlap suppression quantitatively.
- Double-exponential convergence: Elegant link between guidance strength and diffusion rate.
- Bridges gaps: Unifies discrete and continuous CFG theories.
- Empirical alignment: Simulations verify analytical predictions.

**Weaknesses:**

- Heavy reliance on exact score and continuous-time limit; numerical approximations and learned scores are not analyzed.
- Empirical validation is illustrative rather than large-scale.
- Some proofs deferred to appendices could benefit from intuitive discussion in the main text.
- Limited exploration of $D \ge 3$ behavior; higher-dimensional extension remains conjectural.
- Minor presentation complexity (dense notation, multi-index expressions).

**Questions:**

- Could the authors extend the proof techniques to approximate scores (learned $s_\theta$) or noisy simulations?
- How sensitive are the observed phenomena to the choice of the forward process (absorbing vs. uniform)?
- Is the double-exponential convergence rate provably tight, or an upper bound?
- In 2D, does the deviation from the tilted distribution scale polynomially or exponentially in overlap size?
- Could these results suggest a new adaptive guidance schedule where w increases dynamically?
- How might the regional weighting structure $A^{z,w}_i$ inform geometry-aware training or regularization?
- Are there potential connections to discrete optimal transport or entropic regularization frameworks?
- Could the asymptotic results be empirically observed in token-level text diffusion models?
- Are the coefficients $c_x,d_x$ interpretable as marginal reweighting factors for token dependencies?
- How might partial overlapping supports in high dimensions influence generalization or calibration?

---

> ### Author Response · Authors · 2025-11-21
>
> We thank the reviewer for the thoughtful feedback. We address the mentioned weaknesses and questions below.
>
> Regarding **weaknesses**:
>
> For (1) on *heavy reliance on exact score and continuous-time limit*, our focus on exact scores and continuous reverse dynamics follows standard theoretical practice aimed at isolating the intrinsic effect of classifier-free guidance. Importantly, our assumptions on the underlying data distribution are **fully general** and not idealized—they allow arbitrary class-conditional distributions and do not impose any simplifying structural constraints. Regarding *analsysi of numerical errors*, they can be **incorporated directly**: our formulas remain applicable when replacing the exact reverse dynamics with standard numerical solvers such as Euler-Maruyama, and the qualitative behavior we predict persists under practical small step sizes. Regarding *analyzing learned scores*, it depends intricately on **architecture and optimization** and remains poorly understood even in continuous diffusion; while we do not analyze it directly, our results provide geometric intuition for how inductive biases in learned scores may reshape the generated distribution.
>
>
> For (2) on *minimal experiments without robustness or scalability exploration*, this is intentional. The primary goal of the experiments is to validate the specific geometric and distributional effects predicted by the theory. Controlled low-dime settings allow direct comparison with the exact analytical results, and our **5D** and **MNIST** experiments demonstrate that the same qualitative patterns persist in more realistic scenarios. A full-scale empirical study would be valuable but lies outside the scope of a theory-focused paper.
>
> For (3) on *proofs being deferred to the appendix*, we appreciate the suggestion. In the revised manuscript, we will add short intuitive explanations in the main text to clarify the key ideas behind the results while keeping formal proofs in the appendix.
>
> For (4) on *limited exploration of $D\ge 3$*, we agree that extending our explicit characterization beyond $D=2$ is challenging: the algebraic complexity grows rapidly, and obtaining closed-form expressions in higher dimensions is notoriously difficult even for continuous diffusion with CFG (see [1]). Our work should therefore be viewed as the **first explicit and rigorous analysis** in the masked discrete setting, with higher-dimensional extensions left for future work.
>
> For (5) on *dense notation and multi-index expressions*, this is largely unavoidable due to the interacting quantities in the reverse dynamics, especially in 2D. In the revised draft, we will streamline notation and add a short notation table in the appendix to improve readability.

---

> > ### Author Response · Authors · 2025-11-21
> >
> > Regarding **questions**:
> >
> > For Q(1), please see our response for weakness (1).
> >
> > For Q(2), our **quantitative** results rely on the structured sparse reverse rate matrices in masked diffusion, hence do not directly extend to D3PM-uniform. However, the underlying **qualitative** mechanisms that *guidance amplifies class-private regions and suppressing shared ones* should persist because they arise from the mixture structure and the CFG tilting step rather than the absorbing kernel itself.
> >
> > For Q(3), the double-exponential rate is tight: in 1D the $\mathrm{TV}$ expression is exact, and in 2D Proposition 3.4 gives upper and lower bounds with matching double-exponential dependence on the guidance strength $w$.
> >
> > For Q(4), in 2D the deviation from the tilted distribution depends on local marginal ratios rather than a single global overlap size. In uniform examples, these ratios correspond to overlapping-set cardinalities, but in general distributions the deviation is governed by coordinatewise dominance rather than global overlap.
> >
> > For Q(5), our results do suggest the possibility of adaptive or time-varying guidance schedules. Since our analysis separates how $w$ shapes the distribution from how it accelerates convergence, exploring time-dependent guidance is a natural extension, though we do not pursue it here.
> >
> > For Q(6), the regional weights $A_i^{z,w}$ show how guidance shifts probability toward class-private regions, suggesting a natural form of geometry-aware regularization: adjust the training objective so that the learned score’s inductive bias **counteracts** this generation bias. One could, for example, introduce a curriculum-style loss weighting that reflects how different regions contribute under a given guidance strength $w$, encouraging the score network to correct for the over-amplification of private regions during guided sampling. While we do not explore this direction here, the structure of $A_i^{z,w}$ provides a natural foundation for such regularization.
> >
> > For Q(7), the connections to discrete optimal transport and entropic regularization are conceptual: guidance redistributes probability in a way reminiscent of transport plans becoming sharper as entropy decreases. However, CFG in masked discrete diffusion does not explicitly solve an OT problem, and we make no such formal claim.
> >
> > For Q(8), analogous asymptotic behavior may arise in token-level text diffusion models, though verifying this would require careful experimentation. The high dimensionality and structured nature of text prevent our exact formulas from carrying over, but the qualitative effects we identify may still be observable, for example, through shifts in token marginals or reduced variation in semantically uncertain positions under guidance. While we do not study text models here, our theory suggests that similar asymptotic patterns could appear when the model and prompt induce structured overlaps analogous to those in our low-dim framework.
> >
> > For Q(9), the coefficients $c_{x_1}, d_{x_2}$ are indeed interpretable as marginal reweighting factors. Each of $c_{x_1}$ and $d_{x_2}$ is like a generalization of the coefficient $\mathcal{Z}$ in 1D: for example, $c_{x_1}$ measures the difference between the class distribution and the full data distribution through conditioning on fixed first coordinate $x_1$.  These conditioned marginal preferences enter the reverse transition rates and determine which directions of movement are encouraged after guidance is applied. They therefore capture different steering effects along different directions, which does not arise in 1D.
> >
> > For Q(10), guidance systematically shifts probability away from overlapping regions and toward class-private ones. This can impact **generalization**, by reducing mass on valid but ambiguous samples, and **calibration**, by sharpening predictions in uncertain regions. The final effect depends jointly on this intrinsic generation bias and the inductive biases introduced during score learning.

---

### Official Review · Reviewer_9vfg · 2025-11-01

**Soundness:** 3
**Presentation:** 2
**Contribution:** 2
**Rating:** 6
**Confidence:** 4

**Summary:**

This paper extend the analysis of CFG in continuous state diffusion to the masked discrete state diffusion. It has two main results:

* In 1d situation, for masked discrete state diffusion, with direct construct of \\(\hat{Q}\\), we can reach the tilted distribution.
* In 2d situations, it is not that simple.
* This paper also proposes some analysis on multi-guide situation.

**Strengths:**

* The theoretical analysis and calculations are solid, especially the ability to derive exact distribution results— outperforming continuous-state diffusion on CFG in the 1D case.
* The inclusion of the multi-guidance setting adds depth and richness to the paper.

**Weaknesses:**

* Unfortunately, the 1D setting is overly simplistic. While the proposed techniques are effective in 1D, they become increasingly complex in * 2D and are difficult to generalize to higher dimensions due to inherent limitations.
* The experimental evaluation relies too heavily on toy examples, which weakens the practical impact of the work.

**Questions:**

n/a

---

> ### Author Response · Authors · 2025-11-21
>
> We thank the reviewer for the thoughtful feedback. We respond to the mentioned weaknesses below.
>
> >(1) Unfortunately, the 1D setting is overly simplistic. While the proposed techniques are effective in 1D, they become increasingly complex in * 2D and are difficult to generalize to higher dimensions due to inherent limitations.
>
> We agree that the 1D case is intentionally simplified. Its role in the paper is not to suggest direct generalization to high-dim, but to provide a **clean baseline** that highlights how dramatically the behavior of masked discrete diffusion with CFG changes once we move beyond 1D: in 1D, guidance acts as a simple exponential tilt, and the generated distribution matches this tilt exactly. This transparency helps reveal that the 2D case introduces genuinely new phenomena—marginal terms, coordinate interactions, and region-specific weighting, that have no analogue in 1D. On the comment *2D case is overly complex and difficult to generalize to high-dim*, the 2D analysis is **necessarily more involved** because it must encode interactions between coordinates and richer overlap structures, yet our results show that meaningful and explicit structure can still be extracted. Extending the same level of exactness to high-dim is difficult due to the rapid growth in marginal interactions and overlap patterns, but the 2D case already illustrates the central message: once multiple coordinates interact, masked discrete diffusion with CFG exhibits fundamentally richer and more anisotropic behavior than the 1D baseline.
>
> >(2) The experimental evaluation relies too heavily on toy examples, which weakens the practical impact of the work.
>
> Our experiments are focused on controlled settings that allow us to directly **validate the theoretical predictions** and **isolate the geometric effects of guidance**. These effects are clearest in low-dimensional examples where ground-truth structure is fully observable. To complement these toy settings, we also include experiments in **five dimensions** and on **MNIST**, which demonstrate that the qualitative behaviors predicted by our theory persist in more realistic scenarios. While an extensive large-scale empirical study is beyond the scope of a theory-driven paper, we believe the current experiments effectively demonstrate the practical relevance of the mechanisms uncovered by our analysis.

---

### Official Review · Reviewer_XSKv · 2025-11-01

**Soundness:** 3
**Presentation:** 2
**Contribution:** 3
**Rating:** 6
**Confidence:** 2

**Summary:**

This paper provides a rigorous analysis of classifier-free guidance (CFG) for masked discrete diffusion. Under exact scores and exact reverse dynamics, the authors derive closed-form reverse dynamics and generated distributions in 1D and 2D. In 1D, CFG precisely samples from the tilted distribution pz,w, whereas in 2D, the generated distribution deviates with explicit marginal dependent reweighting, shifting mass from overlapping to class-specific regions. For large guidance strength $w$, the total variation (TV) to the terminal distribution decays in time with a rate that is double-exponential in $w$. The paper features experiments on synthetic 1D, 2D and higher-dimensional toy setups, plus MNIST case studies, which qualitatively support the theory.

**Strengths:**

(i) The paper provides the first rigorous treatment of CFG in discrete masked diffusion with explicit formulas in 1D and 2D

(ii) The paper provides clear geometric interpretation. Guidance suppresses overlapping regions and amplifies "private" regions, quantified via region-wise weights in 2D

(iii) The paper's simple, targeted experiments align with theory

**Weaknesses:**

(i) The paper's scope appears limited to masked absorbing diffusion and low dimensions, an extension to higher dimensionality $D > 2$ remains informal

(ii) The paper poses idealized assumptions (exact scores, exact reverse simulation) with little analysis of approximation / discretization error or robustness under practical solvers

(iii) While the main contribution of this work is a theoretical discussion, the scope of experiments remains thin, and the robustness and scalability of results is underexplored

**Questions:**

In addition to the weaknesses outlined in points (i-iii), I present the following questions for the authors to address:

(1) Proposition 3.3 partitions the space into regions $\mathcal{R}_1, \ldots, \mathcal{R}_4$. Can you provide examples of real data where such region decompositions would be meaningful?

(2) The analysis assumes exact concrete scores. How would score approximation errors change the conclusions?

(3) The findings are specific to masked discrete diffusion, following the absorbing forward process. Would the conclusions still hold for other discrete processes (e.g., D3PM-uniform modelling)?

(4) Strong guidance suppresses shared regions and reduces sample diversity. Is this “loss of diversity” always undesirable, or can it be beneficial in certain applications?

(5) What are the main obstacles to extending the exact analysis beyond 2D, and what aspects of the results are most likely to generalize?

(6) In 2D you introduce marginal coefficients $\lbrace c_x, d_x \rbrace_{1 \leq x \leq N}$ that encode the steering effect of guidance on marginals, consequently influencing the sampling dynamics. What intuition can we build for how these coefficients arise and what they represent?

(7) [minor concern w.r.t. to presentation] Figure 2 feels slightly out of place, can you place it at the top of page 8?

(8) [minor concern w.r.t. to presentation] While Notation was shortly introduced in the preliminaries section, could you provide a simple table of notations in the appendix to further improve overall readability and accessibility of your work?

---

> ### Author Response · Authors · 2025-11-21
>
> We thank the reviewer for the thoughtful feedback. We address the mentioned weaknesses and questions below.
>
> Regarding **weaknesses**:
>
> For (i) on *limited scope to masked diffusion and low-dim*, we agree that extending our explicit characterization beyond $D=2$ is challenging: the algebraic complexity grows rapidly, and obtaining closed-form expressions in higher dimensions is notoriously difficult even for continuous diffusion with CFG (see [1]). Our work should therefore be viewed as the **first explicit and rigorous analysis** in the masked discrete setting, with higher-dimensional extensions left for future work.
>
> [1]: Chidambaram, Muthu, et al. "What does guidance do? a fine-grained analysis in a simple setting." Advances in Neural Information Processing Systems 37 (2024).
>
> For (ii) on *idealized assumptions without analyzing discretization/score-approximation error*, our focus on exact scores and exact reverse dynamics follows standard theoretical practice aimed at isolating the intrinsic effect of classifier-free guidance. Importantly, our assumptions on the underlying data distribution are **fully general**, allowing arbitrary class-conditional distributions and do not impose any simplifying structural constraints. **Discretization error can be incorporated directly**: our formulas remain applicable when replacing the exact reverse dynamics with standard numerical solvers such as Euler-Maruyama, and the qualitative behavior we predict persists under practical small step sizes. **Score approximation error depends intricately on architecture and optimization** and remains poorly understood even in continuous diffusion; while we do not analyze it directly, our results provide geometric intuition for how inductive biases in learned scores may reshape the generated distribution.
>
>
> For (iii) on *minimal experiments without robustness or scalability exploration*, We agree that the primary contribution of this work is theoretical. The experiments serve to illustrate the core mechanisms identified by our analysis. The behaviors we highlight (e.g., how guidance reallocates mass across regions) consistently appear across settings broader than those shown. Extending experiments to larger models and higher-dimensional tasks is an important direction enabled by the theoretical foundation we provide.

---

> > ### Author Response · Authors · 2025-11-21
> >
> > Regrading **questions**:
> >
> > For (1), the region decomposition in Proposition 3.3 aligns with structures seen in real data, such as protein sequences [2], where conserved, partially conserved, and flexible positions naturally map to $R_1$, $R_{2,i}/R_3$ and $R_4$.
> >
> > [2]: Gruver, Nate, et al. "Protein design with guided discrete diffusion." Advances in neural information processing systems 36 (2023)
> >
> > For (2), score approximation error interacts with architecture and optimization and is difficult to characterize; empirically, the fact that moderate guidance is optimal may suggest that score approximation error in training may partially offset the strong reshaping predicted by our theory.
> >
> > For (3), our quantitative results rely on the absorbing structure of masked diffusion, hence do not directly extend to D3PM-uniform. However, the underlying qualitative mechanisms that *guidance amplifies class-private regions and suppressing shared ones* should persist because they arise from the mixture structure and the CFG tilting step rather than the absorbing kernel itself.
> >
> >
> > For (4), when the goal is highly class-specific generation (e.g., producing protein sequences characteristic of one family), concentrating mass on class-unique regions can improve fidelity. When broad coverage within class is desired, very strong guidance may oversuppress valid variations. Our analysis makes this tradeoff explicit.
> >
> >
> > For (5), the main obstacle to extending the exact analysis beyond 2D is the rapid increase in algebraic and combinatorial complexity: the reverse rate matrix involves many interacting marginal and conditional quantities, and the structure of class overlaps becomes much richer. These factors make it difficult to obtain a closed form characterization in high-dim. However, the underlying mechanisms revealed by our analysis are not tied to low-dim: the mixture structure in Assumption 1.1 and the tilting step in CFG remain the same in high-dim, and they drive key qualitative effects such as the suppression of states that overlap with other classes and the amplification of states that are specific to the target class. The stronger dependence of the convergence rate on the guidance strength also comes from these general ingredients. For these reasons we expect similar structural patterns, and possibly even analogous formulas, to hold in high-dim, although making them explicit remains challenging.
> >
> > For (6), The coefficients $c_{x_1}$ and $d_{x_2}$ appear because, in 2D, guidance affects the joint probabilities through the marginal distributions along coordinates. Each of $c_{x_1}$ and $d_{x_2}$ is like a generalization of the coefficient $\mathcal{Z}$ in 1D: for example, $c_{x_1}$ measures the difference between the class distribution and the full data distribution through conditioning on fixed first coordinate $x_1$.  These conditioned marginal preferences enter the reverse transition rates and determine which directions of movement are encouraged after guidance is applied. They therefore capture different steering effects along different directions, which does not arise in 1D.
> >
> > For (7) and (8), we appreciate the suggestions, and we will make the recommended changes in the revised manuscript.

---

### Meta-Review · Area_Chair_Aitr · 2026-01-05

**Summary:**

This paper presents an analysis of classifier-free guidance (CFG) in masked discrete diffusion models of 1D and 2D distributions. The paper is nearly entirely theoretical and finds not-surprising results that guidance concentrates on the specified class (amongst other things).

The reviewers broadly agreed about the correctness of the paper and uniformly noted its limitations w.r.t. to dimension.

There was one particularly strong review that I discount because it had no real content nor showed any real engagement with the paper.

My personal reading of the paper is that it is a very borderline at best paper, not because it is wrong or badly written, simply because it utterly fails to communicate the value of the work.  It is a "we found something that we probably knew existed" kind of paper that probably will get in largely because it has a bunch of math in it.

**Reviewer Concerns:**

The reviewers noted the low-dimensionality of the problems considered.  Learned score approximation error and large-scale empirical validation were also noted as being lacking.

**Reviewer Scores:**

I don't think reviewer scores would have changed much if at all.

---

### Decision · Program_Chairs · 2026-01-26

Accept (Poster)